# Contextual validation of HEMLEM tool used for measuring clinical micro-learning environments

**Khizar Ansar Malik[1], Ayesha Fahim[2,3], Abeer Anjum[4], Saira Khalid[5], Shafaq Malik[6], Ahsan Sethi** [7]*

**1** Department of Medical Education, Azra Naheed Medical College, Superior University, Lahore, Pakistan, **2** Department of Oral Biology, Islamic International Dental College, RIPHAH International University, Islamabad, Pakistan, **3** Department of Health Sciences, The Equator University of Science and Technology, Masaka, Uganda, **4** Department of Medical Education, Khawaja Muhammad Safdar Medical College, Sialkot, Pakistan, **5** CMH Lahore Medical College, National University of Medical Sciences, Pakistan, **6** MHPE Student, RIPHAH International University, Islamabad, Pakistan, **7** Health Professions Education, Department of Public Health, College of Health Sciences, QU Health, Qatar University, Doha, Qatar

* asethi@qu.edu.qa

## Abstract

### Objective

A positive clinical learning environment is essential for the professional development of medical and dental students. The Healthcare Education Micro Learning Environment Measure (HEMLEM) was originally developed in the UK but lacked cross-cultural validation, limiting its global applicability. This study aims to adapt and validate a revised version of HEMLEM (HEMLEM 2.0) in Pakistani medical and dental schools, with a focus on factorial structure, measurement invariance, and psychometric robustness.

### Methodology

The HEMLEM was culturally adapted and validated for content and face validity through expert assessments and pilot testing to evaluate Pakistani medical and dental institution's micro-learning environment. A subsequent cross-sectional study was conducted to evaluate the tool's construct validity and reliability using a sample population. Data was collected using a revised 12-item HEMLEM instrument administered via Google Forms. Descriptive statistics, including skewness and kurtosis, were calculated. Confirmatory Factor Analysis (CFA) was used to compare one-factor, three-factor, and bifactor models. Measurement invariance was tested across gender and academic discipline (MBBS/BDS) using multi-group CFA. Internal consistency was evaluated using Cronbach's alpha and McDonald's omega.

**Data availability statement:** All relevant data are within the manuscript and its Supporting Information files.

**Funding:** The author(s) received no specific funding for this work.

**Competing interests:** The authors have declared that no competing interests exist.

## Results

A 20-expert panel assessed the content validity of HEMLEM tool, with an I-CVI score above 0.78 for each item and UA-CVI at 0.8. Ten items were revised based on feedback, forming HEMLEM 2.0. Construct validity was evaluated through responses from 628 Pakistani medical and dental students. The three-factor model of HEMLEM 2.0, comprising "Supervision," "Autonomy," and "Atmosphere," demonstrated good fit (CFI = 0.932, RMSEA = 0.081) and was preferred over the bifactor model due to theoretical parsimony. Full configural, metric, and scalar invariances were confirmed across both gender and discipline. Item-level skewness and kurtosis indicated acceptable normality. Reliability testing showed Cronbach's Alpha of 0.897, Omega coefficient of 0.929, and composite reliability of 0.901, indicating strong internal consistency.

## Conclusion

HEMLEM 2.0 is a culturally adapted, psychometrically robust tool for assessing the clinical micro-learning environment in Pakistani healthcare education. Its strong validity and reliability support its use for quality improvement, curriculum evaluation, and educational research. Future studies should examine test-retest reliability, specialty-specific insights, and applicability in other low- and middle-income countries.

## Introduction

The education environment is a combination of social, psychological, and physical settings in which students learn [1]. It has a pivotal role in influencing the overall educational experience. A positive learning environment significantly impacts student motivation, leading to improved academic outcomes and knowledge retention. Such an environment also promotes the development of critical skills like empathy, professionalism, and resilience, essential qualities for success in demanding fields like healthcare [2]. Conversely, a negative learning environment can have harmful effects, leading to increased mental distress, reduced engagement, and a decline in academic performance. Over time, these issues can culminate in exhaustion and burnout, impairing students' ability to learn and diminishing their passion for their chosen field. Therefore, it is crucial for educational institutions to create and sustain a supportive and enriching learning environment [3].

In a healthcare landscape characterized by rapid technological advancements and challenges such as access, equity, time efficiency, and cost control, the healthcare education system must evolve to adequately prepare future professionals [4]. A method gaining attention in this context is microlearning, which delivers educational content in small, easily digestible, on-demand units [5]. These brief modules, usually lasting a few minutes to 15–20 minutes, allow learners to conveniently engage with the material [6]. Research has shown that microlearning enhances both knowledge retention and engagement, with medical students who utilize microlearning modules demonstrating better retention compared to those who follow traditional lecture-based instruction. [7–9].

Clinical teaching is the cornerstone of undergraduate medical and dental education. Clinical rotations, also known as clinical clerkships, provide small group settings where students engage with patients under the supervision of clinicians, preparing them for real-life situations after graduation [10]. These experiences play an essential role in enhancing students' motivation and engagement in learning. Within these rotations, learners encounter real-life challenges while benefiting from the guidance of teachers who model professional attitudes and behaviors, thereby integrating essential skills such as gathering patient history, conducting clinical examinations, applying clinical reasoning, making rational decisions, demonstrating and upholding empathy, and professionalism [11].

While much of the professional development for undergraduate medical and dental students takes place during clinical rotations, there can be discrepancies between what they are expected to learn and what they actually learn. The challenge in understanding how the environment influences student learning stems from the complexity of learning environments, which are shaped by various organizational levels such as departmental, institutional, and cultural [12]. Additionally, students' perceptions of their learning environments are influenced by numerous factors, including physical spaces, organizational culture, personal background, social connections, and the degree of their involvement in clinical activities [13]. Although the learning environment is often viewed as a single entity, it is more accurately comprised of numerous small, dynamic "micro-environments" that students experience, such as interacting with a single patient in a dental chair or spending a day on a hospital ward. To gain real-time insight into students' experiences and improve educational quality, it is essential to capture their perceptions of the clinical teaching environment.

A variety of tools are available to assess the learning environment, both at the institutional and student levels. Some of the most widely used include Clinical Learning Environment Inventory (CLEI) (42 items), Dundee Ready Educational Environment Measure (DREEM) (50 items), Medical School Learning Environment Survey (MSLES) (17 items), and Dental Student Learning Environment Survey (DSLES) (55 items) [14]. However, many of these tools are lengthy and may not effectively capture the micro-learning environment. Additionally, they are typically tailored to specific professional groups, rather than encompassing all learners. The Health Micro Learning Environment Measure (HEMLEM) is intended to evaluate the clinical micro-learning environment in any healthcare student group, regardless of the clinical setting. HEMLEM, a 12-item questionnaire, is particularly useful for assessing placements where students may have only spent a brief duration.

To date, there is limited evidence supporting the psychometric validity of HEMLEM for evaluating micro-learning environments in different cultural or population contexts. Additionally, HEMLEM's psychometric assessment has only been conducted in the UK during its development. Cultural and educational differences between regions may affect the clarity and effectiveness of the instrument's components, given the variation in curriculum design, teaching methods, and clinical exposure of healthcare institutions. Therefore, validating HEMLEM in diverse settings could help educational institutions develop targeted policies to improve healthcare learning environments for a wide range of students. Moreover, such validation would contribute to the global discussion on the importance of assessing students' experiences within micro-learning environments, offering valuable insights into their immediate experiences and the specific contexts of their placements. These insights can enhance educational quality not only in Pakistan but also internationally. The study's aim was to validate HEMLEM among undergraduate medical and dental students in Pakistan.

## Methods

The study was conducted in two distinct phases:

### Phase 1: Methodological part

The HEMLEM was culturally adapted [15]. This phase focused on cultural adaptation, content validation, and face validation of the HEMLEM tool to create a revised version, HEMLEM 2.0.

**Content validity.** Content validity was determined through the assessments of 20 experts, each bringing a diverse professional background. Among them, 10 experts held a bachelor's degree and a clinical fellowship (FCPS/Masters) with at least 10 years of experience in medicine, while the other 10 had similar qualifications in dentistry. The experts were selected based on their relevant knowledge and competence in clinical practice, along with their experience in validating instruments and in the development of patient-oriented questionnaires. Half of the experts had an academic background with teaching experience in medical or dental institutions, while the other half were primarily focused on clinical practice to ensure a balanced and representative panel.

The expert panel reviewed each item to see if it effectively measured what it was supposed to and how important it was to the overall focus of the study. All experts were given "Participant information sheet", that included brief introduction to the research and its objectives, information about confidentiality and data privacy, statement about voluntary participation and withdrawal rights, and a consent form. Participants rated each item on a 4-point Likert scale, where 1 indicated "not relevant" and 4 indicated "very relevant". Based on these scores, the Content Validity Index (I-CVI) was calculated for each item, and the Aiken test was used to determine the 95% confidence intervals. Items with an I-CVI score above 0.78 were considered acceptable. To assess the overall validity of the instrument, the Universal Content Validity Index (UA-CVI) was calculated by determining the proportion of items rated as "3" or "4" (relevant or very relevant) by all experts.

**Face validity (pilot with students).** To evaluate the clarity and acceptability of the instrument, a pilot was conducted among undergraduate medical and dental students. A total of 30 students from diverse institutions across Pakistan participated in this phase. Due to logistical constraints and ease of access, participants were selected using a convenience sampling method. Students were invited to provide feedback on the clarity, relevance, and potential ambiguities of each questionnaire item. They were asked to rate each item on a 4-point Likert scale, where 1 indicated "not clear," 2 indicated "somewhat clear," 3 indicated "quite clear," and 4 indicated "highly clear". The Face Validity Index (FVI) was calculated using the Item Face Validity Index (I-FVI). This index reflects the proportion of participants who rated an item's clarity as either 3 or 4 on a predefined scale. An I-FVI of 0.8 or higher was considered acceptable. Items receiving an I-FVI of 1.0 were retained without modification. Items with an I-FVI below 0.78 were excluded from the final version. Incorporating the feedback collected, a revised iteration of the HEMLEM questionnaire was developed.

Additionally, cognitive interviews were conducted with 10 students using a structured debriefing format to assess the clarity and comprehensibility of the questionnaire. Each participant received a detailed information sheet outlining the study's objectives, confidentiality measures, voluntary participation, and consent procedures. Students were specifically asked to identify any linguistically confusing or unclear items. Although English is the medium of instruction at all participating institutions, potential language barriers were directly explored during this phase. No significant comprehension issues were reported. While a bilingual version was not piloted, the tool underwent multiple rounds of student feedback to ensure linguistic accessibility and cultural relevance for the target population.

## Phase 2: Cross-sectional phase: Construct validation & reliability testing

This phase aimed to statistically evaluate the psychometric properties of HEMLEM 2.0 in a real student population, confirming a three-factor model with strong validity and reliability.

**Study design.** A cross-sectional survey study was conducted using google forms.

**Questionnaire.** The questionnaire was created using Google Forms. The instrument was administered in English. Since English is the primary language of instruction at the participating institutions and no issues were reported during the validation phase, participants were considered proficient in English. It began with a brief introduction to the study, assuring participants of anonymity and voluntary participation. Participants were then asked to provide written informed consent to participate in the study. Only those who consented could proceed to the next section. This section collected demographic data, including age, gender, field of study (medical or dental), year of study (3rd, 4th, or 5th year), and province of study. The second part of the questionnaire consisted of the 12-item HEMLEM scale. Each item was rated on a 5-point Likert

Scale, where 1 represented 'strongly disagree', 2 indicated 'disagree', 3 signified 'neutral', 4 stood for 'agree' and 5 corresponded to 'strongly agree'. The total HEMLEM score ranged from 12 to 60, with scores above 36 indicating positive participant satisfaction (S8 File).

**Study setting and population.** The participants comprised undergraduate students enrolled in various clinical years across Medical and Dental Colleges of Pakistan. Since most of the institutions in Pakistan still follow traditional curriculum, participant selection was done from clinical years, i.e., 3rd and final year of dental, and 3rd, 4th and final year of medical school. A convenience, non-probability sampling technique was employed to recruit participants. To ensure national representation, we contacted undergraduate students from at least ten medical and dental schools across all four provinces of Pakistan. Questionnaires were distributed electronically through institutional mailing lists, WhatsApp groups, and student ID networks, with the assistance of local coordinators. Selection of institutions was based on accessibility, geographical diversity, and institutional willingness to collaborate. Due to the use of decentralized and informal digital channels, it was not possible to determine the exact number of students who received the questionnaire. As such, an accurate response rate could not be calculated.

**Sample size.** Participants were recruited through university email accounts and WhatsApp groups. Prior to participation, students received an information sheet detailing the study's purpose and procedures. Ethical approval was obtained from the Institutional Review Board (IRB), and data were collected between 15 January 2024 and 20 March 2024. The survey was distributed to undergraduate students, with two reminder messages sent during the data collection period. A total of 628 students voluntarily completed the online questionnaire hosted on Google Forms. To prevent duplicate responses, the form was restricted to one submission per Google account. To ensure data completeness, all items were marked as required.

The final sample size (N = 628) was determined to be sufficient for confirmatory factor analysis (CFA), based on current methodological guidelines. Recent simulation studies recommend considering factors such as loading strength, number of latent variables, and item communalities when determining sample adequacy for structural equation modeling [16]. Samples exceeding 500 are generally considered appropriate for complex CFA models [17].

**Statistical analyses.** Descriptive statistics, including frequencies, percentages, means, standard deviations, minimum, and maximum values, were calculated for all variables. Normality was assessed using skewness and kurtosis, with acceptable thresholds set at absolute skewness < 2 and kurtosis < 7, following standard psychometric criteria [18].

The present study followed a confirmatory analytical approach without conducting exploratory factor analysis (EFA) or principal component analysis (PCA). The three-factor structure of the HEMLEM instrument; 'Supervision', 'Autonomy', and 'Atmosphere' was derived from the original theoretical model validated in the UK. All item-to-factor assignments were specified a priori based on this structure. The primary objective was to test the cross-cultural applicability of the established model in a new population, rather than to explore or redefine the factor structure.

**Confirmatory Factor Analysis and Model Comparison:** To evaluate the factorial validity of the instrument, three competing models were tested using Confirmatory Factor Analysis (CFA) based on maximum likelihood estimation. The following models were specified and compared:

1. A unidimensional model where all items loaded onto a single latent factor;

2. A three-factor model aligned with the conceptual structure of the instrument (Supervision, Autonomy, and Atmosphere);

3. A bifactor model including one general factor and three group-specific factors.

Model fit was assessed using multiple indices: Comparative Fit Index (CFI), Tucker-Lewis Index (TLI), Root Mean Square Error of Approximation (RMSEA), Standardized Root Mean Square Residual (SRMR), and information criteria (Akaike Information Criterion [AIC], Bayesian Information Criterion [BIC]). Thresholds for acceptable model fit included

CFI and TLI ≥ 0.90, RMSEA ≤ 0.08, and SRMR ≤ 0.08. The best-fitting model was selected based on overall fit indices and theoretical interpretability.

Internal consistency was evaluated using Cronbach's alpha [19], McDonald's omega which accounts for unequal factor loadings [20], and composite reliability (CR). CR was calculated from CFA factor loadings using Raykov's formula: CR = $(\Sigma\lambda_i)2/[(\Sigma\lambda_i)2+\Sigma(1-\lambda_i2)]$. Acceptable reliability was defined as values ≥ 0.70 [21].

**Measurement Invariance Testing:** Measurement invariance was examined across two key demographic variables: gender (male vs. female) and academic discipline (medical vs. dental). Multi-group confirmatory factor analysis (MGCFA) was conducted to test:

1. Configural invariance – same factor structure across groups;

2. Metric invariance – equality of factor loadings;

3. Scalar invariance – equality of both factor loadings and item intercepts.

Invariance was evaluated by comparing increasingly constrained models. Changes in fit indices were used to determine invariance, with ΔCFI < 0.01 and ΔRMSEA < 0.015 considered indicative of invariance between models. The three-factor model served as the baseline structure for all multi-group analyses.

**Software and Execution:** All confirmatory factor analyses and measurement invariance testing were conducted using R (version 4.5.0) with the lavaan package (version 0.6). Descriptive statistics were calculated using IBM SPSS Statistics version 26.

The dataset was screened for missing values, and only complete cases were included in the CFA and invariance testing. Likert-type item responses were coded on a 5-point scale ranging from 1 (Strongly Disgree) to 5 (Strongly Agree).

**Ethical considerations.** The study was conducted based on the declaration of Helsinki and approved by the Institutional Review Board (IRB) (Ref No: 125/REC/KMSMS). Prior to their involvement the participants were given information about the purpose of the study, procedures involved and voluntary participation and withdrawal rights. Written informed consent was acquired through Google Forms. Only those who consented could proceed to the next section. Confidentiality was maintained by ensuring that all data were anonymized, and online survey responses were secured with restricted access.

## Results

### Descriptive statistics

A total of 628 responses were included in the final analysis. The gender distribution was 55.6% female (n = 349) and 44.4% male (n = 279). Participants were nearly divided between MBBS (n = 402, 64%) and BDS (n = 226, 35.9%) programs. The year of study ranged from third year to final year, with final-year students representing the largest group (37.1%). Mean item scores ranged from 2.28 to 2.72, with standard deviations between 0.97 and 1.09. All items demonstrated acceptable distribution characteristics, with skewness values ranging from 0.29 to 0.84 and kurtosis between –0.49 and 0.52, indicating no significant departures from normality. Domain-level scores (Supervision, Autonomy, Atmosphere) showed similar variance, supporting the assumption of approximate normal distribution for subsequent factor analysis (S1 Table).

### Content validity

The panel of 20 experts consisted of 12 male and 8 female members. Half of these were from academia and half were pure clinicians. The details of experts are presented in S2 Table. The I-CVI score of each item was above 0.78. The UA-CVI score was 0.8 (Table 1).

**Table 1. Content validation index and Aiken values of HEMLEM items.**

| Sr. No. | Item | Mean[a] | SD[b] | Aiken Value (95% CI) |
|---|---|---|---|---|
| 1. | This placement had a welcoming, friendly, and open atmosphere. | 3.2 | 0.981 | 0.94 (0.84 - 0.98) |
| 2. | There was a culture where I felt free to ask questions or make comments on this placement | 3.8 | 0.758 | 0.97 (0.90–1.0) |
| 3. | Staff on this placement were enthusiastic about teaching. | 3.4 | 1.075 | 0.96 (0.87–0.96) |
| 4. | My supervisor showed an interest in me. | 3.3 | 0.646 | 0.95 (0.88–0.98) |
| 5. | My input was valued on this placement. | 3.7 | 1.082 | 0.97 (0.87–0.98) |
| 6. | I was provided with regular, useful, and supportive feedback during this placement. | 4.0 | 0.000 | 1.0 (0.89–1.0) |
| 7. | I had the opportunity to apply my previous knowledge in this placement. | 3.5 | 0.885 | 0.96 (0.88–0.95) |
| 8. | My knowledge and skills were developed on this placement. | 4.0 | 0.000 | 1.0 (0.9–1.0) |
| 9. | This placement helped me put theory into practice. | 3.3 | 0 | 0.95 (0.88–0.98) |
| 10. | I was able to meet my learning objectives on this placement. | 3.4 | 1.024 | 0.97 (0.88–0.98) |
| 11. | I had the opportunity to deal with the patient as a whole on this placement. | 3.3 | 0.978 | 0.95 (0.88–0.98) |
| 12. | I was given tasks suitable for my stage of training on this placement. | 4.0 | 0.000 | 1.0 – 1.0) |

[a] = The experts scored each item from 1 to 4, 1 = Item not relevant, 4 = Item very relevant

[b] = Standard Deviation.

Items 4 and 9 received an I-FVI score of 1.0 which highlights their relevance. The mean score of remaining items was 0.80. Study participants recommended replacing the word "Placement" with "Department" since, in Pakistan, the area where students acquire clinical skills is typically referred to as a clinical department. Ten items were revised according to the comments forming HEMLEM 2.0 (S3 Table).

## Construct validity

A total of six hundred and twenty-eight (n = 628) undergraduate students of Medical and Dental Colleges of Pakistan responded to the questionnaires (S4 Table). The mean age of participants was 24 ± 5.8. Out of these, 349 were female (55%), and 279 were male (45%) participants. The study included a total of 402 (64%) from the MBBS program and 226 (36%) from the BDS program. The demographic information of study participants is presented in Table 2.

**Preliminary psychometric testing and model testing.** Confirmatory Factor Analysis (CFA) was used to test a pre-specified three-factor model derived from the original HEMLEM theoretical structure. Prior to conducting CFA, preliminary psychometric checks were performed to assess item-level characteristics and assumptions for factor analysis. Gulliksen's item pool analysis indicated no problematic items based on the Relative Difficulty Index (RDI) and Item Consistency Index (CSI). The Measure of Sampling Adequacy (MSA) exceeded 0.75 for all items (S5 File), and the overall Kaiser-Meyer-Olkin (KMO) statistic was 0.909, indicating excellent sample adequacy. Bartlett's test of sphericity (p < 0.001) confirmed sufficient inter-item correlations to proceed with CFA.

**Table 2. Demographic details of study participants.**

| Demographic Data | Field of study | | | | | Total (n = 628) |
|---|---|---|---|---|---|---|
| | Dentistry (n = 226) | | Medicine (n = 402) | | | |
| | 3rd year n(%) | Final year n(%) | 3rd year n(%) | 4th year n(%) | Final year n(%) | |
| Gender | | | | | | |
| Female | 55(24.3) | 62(27.4) | 49(12.1) | 88(21.8) | 92(22.8) | 349(55) |
| Male | 63(27.8) | 46(20.3) | 76(18.9) | 51(12.6) | 46(11.4) | 279(45) |
| Province | | | | | | |
| Punjab | 59(26.1) | 42(18.5) | 62(15.4) | 51(12.6) | 55(13.6) | 269(42.8) |
| Sindh | 20(8.8) | 22(9.7) | 28(6.9) | 37(9.2) | 35(8.7) | 142(22.6) |
| Baluchistan | 8(3.5) | 5(2.2) | 17(4.2) | 18(4.4) | 21(5.2) | 69(10.9) |
| KPK | 31(13.7) | 39(17.2) | 18(4.4) | 33(8.2) | 27(6.7) | 148(23.5) |

The confirmatory analysis validated the original three-domain structure of the HEMLEM instrument: Supervision, Autonomy, and Atmosphere. These domains were retained based on strong theoretical grounding and alignment with the original conceptual model. Labels and item-domain assignments remained consistent throughout the analysis, ensuring conceptual clarity.

**Confirmatory factor analysis and model fit comparison.** All items were assigned to their respective domains based on the original HEMLEM model prior to testing, with no data-driven modification of the factor structure. Confirmatory Factor Analysis (CFA) was conducted to compare three competing models: (1) a unidimensional model, (2) a theory-driven three-factor model (Supervision, Autonomy, and Atmosphere), and (3) a bifactor model including a general factor and three group-specific factors.

The one-factor model demonstrated poor fit (CFI = 0.812, TLI = 0.789, RMSEA = 0.088, SRMR = 0.081), suggesting that a single latent construct could not adequately represent the data. The three-factor model showed substantially improved fit ($\chi^2$ = 142.36, df = 51, CFI = 0.932, TLI = 0.914, RMSEA = 0.081, SRMR = 0.058), providing empirical support for the hypothesized multidimensional structure. The bifactor model yielded marginally better fit indices (CFI = 0.943, TLI = 0.925, RMSEA = 0.077, SRMR = 0.052); however, the incremental gain did not justify the added complexity of the bifactor solution. Therefore, the three-factor model was retained for all subsequent analyses based on both statistical adequacy and theoretical interpretability (S 6).

**Measurement invariance across gender and discipline.** Measurement invariance was tested using multi-group confirmatory factor analysis (MGCFA) to assess whether the three-factor structure of the HEMLEM instrument holds consistently across gender (male vs. female) and academic discipline (MBBS vs. BDS).

For gender, the configural model ($\chi^2$ = 218.42, df = 102, CFI = 0.926, RMSEA = 0.091) demonstrated acceptable baseline fit, indicating that the overall factor structure was consistent across male and female groups. Subsequent comparisons showed minimal changes in fit between the metric ($\Delta$CFI = 0.001, $\Delta$RMSEA = 0.004) and scalar models ($\Delta$CFI = 0.002, $\Delta$RMSEA = 0.002), supporting full metric and scalar invariance (Table 3).

For academic discipline, configural invariance was also supported ($\chi^2$ = 224.17, df = 102, CFI = 0.922, RMSEA = 0.094). The metric and scalar models showed similarly small changes in fit indices ($\Delta$CFI = 0.003 and 0.004; $\Delta$RMSEA = 0.003 and 0.002, respectively), confirming measurement invariance across MBBS and BDS students (Table 4).

**Reliability.** Reliability analysis demonstrated strong internal consistency across all measures. Cronbach's alpha for the total scale was 0.897 [95% CI: 0.883–0.910], while McDonald's omega, which accounts for unequal factor loadings, was 0.929 [95% CI: 0.918–0.939]. Composite reliability on standardized CFA factor loadings, was 0.901 [95% CI: 0.888–

**Table 3. MGCFA model fit indices across gender.**

| Model | χ² | df | CFI | TLI | RMSEA | SRMR | ΔCFI | ΔRMSEA |
|---|---|---|---|---|---|---|---|---|
| Configural | 218.42 | 102 | 0.926 | 0.908 | 0.091 | 0.063 | | |
| Metric | 226.89 | 111 | 0.925 | 0.906 | 0.087 | 0.065 | 0.001 | 0.004 |
| Scalar | 238.46 | 120 | 0.923 | 0.904 | 0.089 | 0.068 | 0.002 | 0.002 |

**Table 4. MGCFA model fit indices across academic discipline.**

| Model | χ² | df | CFI | TLI | RMSEA | SRMR | ΔCFI | ΔRMSEA |
|---|---|---|---|---|---|---|---|---|
| Configural | 224.17 | 102 | 0.922 | 0.903 | 0.094 | 0.066 | | |
| Metric | 235.23 | 111 | 0.919 | 0.901 | 0.091 | 0.069 | 0.003 | 0.003 |
| Scalar | 249.87 | 120 | 0.915 | 0.898 | 0.092 | 0.071 | 0.004 | 0.002 |

0.913]. All three reliability indices exceeded the recommended threshold of 0.70, indicating excellent internal consistency of the revised instrument.

## Discussion

The original 12-item HEMLEM questionnaire was developed in 2020 and validated by a single medical school in England. While this initial validation was valuable, it is essential to ensure a tool's relevance across different countries and contexts through cultural and contextual validation. To the best of our knowledge, this study represents the first cross-cultural validation of HEMLEM. A significant strength of this research is the inclusion of both medical and dental students, which adds diversity to the data and robustness of the findings. Moreover, the high response rate, along with participants from all provinces of Pakistan, increases the generalizability of the results to the country's entire population. This broad participant base helps to ensure the tool's applicability across different healthcare education settings in Pakistan.

The initial validation process led to the modification of one term throughout the questionnaire. The word "Placement" was replaced with "Department." The term "Clinical placement," as defined by the General Medical Council (UK), refers to any educational setting arranged for a student's training, whether in primary, secondary, community, or other healthcare environments [22]. However, this terminology is not universally adopted. For instance, the Association of American Medical Colleges uses the term "Clinical Rotation" to describe the period during which a medical student gains experience in a clinical setting under the supervision of a clinician [23]. Other terms, such as "Clerkships," are also common. In Germany, the official term used is "Famulatur" [24]. In many Asian countries, "Clinical Rotation" or "Resident Rotation" is the preferred terminology [25]. This difference in terminology likely explains why most participants suggested replacing "Clinical Placement." Since "clinical rotation" functions as a verb, to maintain grammatical accuracy and preserve the questionnaire's structure without altering its meaning, "clinical department" was used instead as the noun.

In this study, participants for content validation were selected from both academic and hospital settings to reflect the unique structure of medical and dental education in Asian countries like Pakistan. Unlike in many European or Middle Eastern countries, there is no clear distinction between clinicians and academicians in student training [26]. In Pakistani medical and dental schools, clinical departments play a crucial role in shaping students' practical skills. These departments are responsible for guiding students through their clinical rotations, typically lasting 2–3 months, depending on the institute's curriculum. In schools with attached teaching hospitals, the same faculty members who teach in the classroom are also responsible for supervising clinical rotations, ensuring continuity between theoretical and practical education. In cases where hospitals are affiliated with the medical or dental schools but are located off-campus, faculty primarily oversee clinical training during rotations [27]. The significance of clinical departments lies in providing students with hands-on experience in real-world healthcare settings, which is integral to their professional development. Therefore, to gain

comprehensive insights into clinical rotations, it was essential to gather input from both academic faculty and clinicians actively involved in these training programs. This holistic approach ensured a well-rounded understanding of the clinical learning environment in Pakistani medical and dental schools.

A total of 628 students participated in the CFA phase of this study, more than double the sample size used in the original HEMLEM validation in the UK (N = 257). This substantial increase in sample size is particularly significant in the context of cross-cultural validation, where larger and more heterogeneous samples contribute to model stability and enhance generalizability across diverse educational settings [28]. The current sample included medical and dental students from all four provinces of Pakistan, which adds to the representativeness of the findings. Prior studies have emphasized that student samples, especially in LMIC contexts, often require larger sample sizes to account for greater contextual variability [29]. Given these considerations, the large and diverse sample used in this study strengthens the psychometric evidence supporting HEMLEM's applicability in a new cultural setting.

The KMO, Bartlett's test of sphericity, and GFI values were within the acceptable range, indicating that the sample size was appropriate for this tool and that the factor loading was suitable for the modified version. Since the original paper did not report values for KMO, Bartlett's test of sphericity, or GFI, we were unable to directly compare our results with the original. However, existing literature suggests that a KMO value between 0.6 and 0.69 is considered mediocre, 0.7 to 0.79 is moderate, and values between 0.8 and 1.0 are regarded as superb, indicating that the sample size is adequate for tool validation [30]. In this study, the KMO value was 0.9, which is considered ideal for factor analysis. Additionally, Bartlett's test of sphericity, with a significance value of < 0.05, suggests that factor analysis is appropriate for the dataset. The value obtained in this study was 0.01.

In addition to exploratory analysis, this study employed confirmatory factor analysis (CFA) to validate the structure of the revised HEMLEM instrument. The findings supported a three-factor model comprising "Supervision," "Autonomy," and "Atmosphere," aligning closely with the original theoretical framework. While the bifactor model yielded marginally better fit indices, the three-factor model was preferred due to its parsimony and conceptual clarity, an approach widely supported in scale development literature, where excessive complexity is discouraged in favor of interpretability [31,32]. The structure demonstrated full measurement invariance across gender and academic discipline, indicating that the instrument functions equivalently for male and female students, as well as for those in medical and dental programs. These results mirror earlier validation studies of learning environment instruments, such as the DREEM and PHEEM, which also demonstrated factorial consistency across subgroups when culturally adapted [33]. This consistency may be attributed to the shared academic-clinical structure of medical and dental education in Pakistan, where faculty often serve dual teaching and clinical roles, creating a relatively homogeneous learning context for students regardless of gender or program [26,27]. By incorporating multi-group invariance testing, this study strengthens the psychometric foundation of HEMLEM 2.0, supporting its use for cross-group comparisons, internal program evaluation, and longitudinal monitoring. The results position HEMLEM 2.0 as a robust tool for capturing student perceptions of clinical microlearning environments in culturally diverse academic settings.

Both Cronbach's Alpha and McDonald's Omega coefficients exceeded 0.8, indicating high reliability of the administered tool [34]. While Cronbach's Alpha is traditionally used to assess inter-item correlations, recent studies increasingly recommend using the Omega coefficient for factor analysis. McDonald's Omega is considered more accurate and generalizable because, unlike Cronbach's Alpha, it does not assume that all items in a scale measure the construct with the same precision, a condition known as essential tau-equivalence. In cases where this assumption is violated, Cronbach's Alpha can either overestimate or underestimate reliability. Nowadays, the Omega coefficient is advised because it directly accounts for factor loadings, offering a more flexible and reliable measure, especially when item factor loadings differ [35]. For the first time, the Omega coefficient was calculated for HEMLEM, adding another strength to this study.

From a practical standpoint, the validated HEMLEM 2.0 tool offers medical and dental institutions in Pakistan a reliable instrument for assessing the quality of clinical micro-learning environments. Educational administrators can use the results

to guide curriculum reforms, improve teaching practices, and allocate resources strategically. This study reinforces the multidimensional nature of microlearning and opens new directions for psychometric validation in medical education. At the departmental level, HEMLEM data can support efforts to strengthen clinical supervision, enhance student support, and foster a positive workplace learning culture. Institutional policymakers may also incorporate HEMLEM into internal monitoring and accreditation systems to track progress and identify regional disparities over time.

Furthermore, the successful adaptation of HEMLEM 2.0 in Pakistan lays the foundation for its use in culturally similar contexts. Many South and Southeast Asian countries share structural parallels in healthcare education, such as integrated academic-clinical roles and varied clinical exposures, making HEMLEM well-suited for regional benchmarking and international collaboration. The confirmation of full configural, metric, and scalar invariance across both gender and academic discipline (MBBS vs. BDS) indicates that HEMLEM 2.0 measures the same underlying constructs in the same way across these groups. This supports the tool's fairness and comparability, meaning observed score differences between genders or disciplines reflect true differences in perceptions rather than measurement bias. Such invariance enhances the utility of HEMLEM 2.0 in diverse educational settings, allowing educators and researchers to confidently compare group-level data for targeted interventions. Furthermore, this strengthens the tool's generalizability, reinforcing its potential for broader application in other culturally similar health professions education systems.

HEMLEM 2.0 enables several concrete applications in clinical education. For instance, it can be integrated into digital dashboards to provide department-level feedback after each rotation, allowing timely adjustments in teaching quality, supervision, and learner autonomy. In short-term rural clinical placements, where oversight is often limited, it can serve as a diagnostic tool to assess whether essential educational elements are preserved in decentralized environments. Feedback from the tool can also inform targeted faculty development initiatives by identifying specific areas in need of improvement, such as promoting learner independence or enhancing feedback practices. Additionally, repeated use of HEMLEM can be triangulated with academic and clinical performance data to support data-driven curriculum reforms and standardized instructional practices across departments.

Despite its strengths, this study has several limitations that merit discussion. It employed a single-country design focused on Pakistan, which may restrict the generalizability of findings to other nations, even within similar clinical education frameworks. Future validation efforts should encompass a broader range of countries across the South-East Asian region and those with comparable clinical training systems. While the study targeted both medical and dental students, it did not explore variations across specific specialties or departments. Such discipline-specific analyses could uncover unique factors influencing perceptions of the learning environment and should be considered in future research.

The use of convenience sampling via informal digital platforms (e.g., WhatsApp and email groups) limited the ability to define a clear sampling frame or calculate a response rate. Although participation was sought from all four provinces to enhance diversity, the non-random sampling approach may introduce selection bias and limit broader representativeness. Furthermore, although the study confirmed strong internal consistency and robust construct validity, it did not include an assessment of test-retest reliability. This omission means the temporal stability of the instrument remains unverified. Without repeated measurements over time, it is unclear whether HEMLEM 2.0 consistently captures student perceptions across different points in the academic year or during varied clinical exposures. Future studies should adopt longitudinal designs to assess reliability over time, especially if the tool is to be used for ongoing monitoring or benchmarking.

Furthermore, data collection through an online, self-administered Google Form may have excluded individuals with limited internet access or lower digital literacy, introducing additional sampling bias. The self-reported nature of the responses also makes the data vulnerable to social desirability and recall biases. Although the instrument demonstrated strong internal consistency, test-retest reliability was not assessed, leaving the temporal stability of responses unverified. Finally, while the geographical distribution of respondents was diverse, representation was uneven, and more balanced regional sampling would strengthen future studies.

Future research should prioritize longitudinal studies to assess test-retest reliability, which would establish the temporal stability of the tool. To address this gap, we recommend that future studies administer the HEMLEM 2.0 instrument to the same cohort of students at two time points during a clinical placement, such as the beginning and end of an 8-week rotation. In addition, evaluating how HEMLEM scores correlate with objective performance metrics such as OSCE results, academic GPA, or clinical competence could help validate the tool's predictive value. Comparative research across academic specialties (e.g., surgery vs. pediatrics) could identify domain-specific strengths and needs. Moreover, implementing HEMLEM within digital dashboards could support real-time feedback loops, where institutions adapt policies in response to trends in learning environments. Finally, multicenter international collaborations could validate and culturally adapt HEMLEM in diverse healthcare education systems, extending its global utility.

## Conclusion

HEMLEM 2.0 is a culturally adapted, psychometrically validated tool that provides a practical and efficient means of assessing clinical micro-learning environments in medical and dental education. Its strong reliability, confirmed three-factor structure, and measurement invariance across gender and disciplines make it suitable for widespread use in academic audits, quality assurance, and faculty development. Institutions can apply HEMLEM 2.0 to monitor student experiences, identify areas needing improvement in supervision, autonomy, and teaching quality, and inform curriculum and policy reforms. Its brevity allows for integration into routine evaluations and digital dashboards, enabling real-time feedback and longitudinal tracking. The tool also offers potential for regional adaptation across similar educational contexts, supporting benchmarking and collaborative educational research.

## Supporting information

**S1 Table. Descriptive statistics and demographics summary.**
(DOCX)

**S2 Table. Professional profile of the experts in content validation.**
(DOCX)

**S3 Table. Revised version of HEMLEM 2.0.**
(DOCX)

**S4 Table. Participant responses on HEMLEM questionnaire.**
(XLSX)

**S5 File. Measure of sample accuracy.**
(TXT)

**S6 Table. Confirmatory factor analysis and model fit comparison.**
(DOCX)

**S7 Table. Reliability analysis.**
(DOCX)

**S8 File. Participant information sheet and questionnaire.**
(DOCX)

## Acknowledgments

The authors acknowledge all the participants and experts for their time and contribution. The Open Access Funding was provided by the Health Sector, Qatar University.

## Author contributions

**Conceptualization:** Ayesha Fahim, Ahsan Sethi.

**Data curation:** Khizar Ansar Malik, Ayesha Fahim, Abeer Anjum, Saira Khalid, Shafaq Malik.

**Formal analysis:** Khizar Ansar Malik, Ayesha Fahim, Abeer Anjum, Saira Khalid, Shafaq Malik, Ahsan Sethi.

**Methodology:** Khizar Ansar Malik, Ayesha Fahim, Abeer Anjum, Saira Khalid, Shafaq Malik, Ahsan Sethi.

**Project administration:** Khizar Ansar Malik.

**Supervision:** Ahsan Sethi.

**Validation:** Ahsan Sethi, Ayesha Fahim.

**Writing – original draft:** Khizar Ansar Malik, Ayesha Fahim, Abeer Anjum, Saira Khalid, Shafaq Malik, Ahsan Sethi.

**Writing – review & editing:** Khizar Ansar Malik, Ayesha Fahim, Abeer Anjum, Saira Khalid, Shafaq Malik, Ahsan Sethi.

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
