## [Decision Letter · Decision Letter 0]

10 Mar 2025

Dear Dr. Sethi,

Thank you for submitting your manuscript to PLOS ONE. After careful consideration, we feel that it has merit but does not fully meet PLOS ONE’s publication criteria as it currently stands. Therefore, we invite you to submit a revised version of the manuscript that addresses the points raised during the review process.

We look forward to receiving your revised manuscript.

Kind regards,

Fatemeh Farshad

Guest Editor

PLOS ONE

**Journal Requirements:**

1. When submitting your revision, we need you to address these additional requirements. Please ensure that your manuscript meets PLOS ONE's style requirements, including those for file naming. The PLOS ONE style templates can be found at https://journals.plos.org/plosone/s/file?id=wjVg/PLOSOne_formatting_sample_main_body.pdf and https://journals.plos.org/plosone/s/file?id=ba62/PLOSOne_formatting_sample_title_authors_affiliations.pdf 2. We note that this data set consists of interview transcripts. Can you please confirm that all participants gave consent for interview transcript to be published? If they DID provide consent for these transcripts to be published, please also confirm that the transcripts do not contain any potentially identifying information (or let us know if the participants consented to having their personal details published and made publicly available). We consider the following details to be identifying information:- Names, nicknames, and initials- Age more specific than round numbers- GPS coordinates, physical addresses, IP addresses, email addresses- Information in small sample sizes (e.g. 40 students from X class in X year at X university)- Specific dates (e.g. visit dates, interview dates)- ID numbers Or, if the participants DID NOT provide consent for these transcripts to be published:- Provide a de-identified version of the data or excerpts of interview responses- Provide information regarding how these transcripts can be accessed by researchers who meet the criteria for access to confidential data, including:a) the grounds for restrictionb) the name of the ethics committee, Institutional Review Board, or third-party organization that is imposing sharing restrictions on the datac) a non-author, institutional point of contact that is able to field data access queries, in the interest of maintaining long-term data accessibility.d) Any relevant data set names, URLs, DOIs, etc. that an independent researcher would need in order to request your minimal data set. For further information on sharing data that contains sensitive participant information, please see: https://journals.plos.org/plosone/s/data-availability#loc-human-research-participant-data-and-other-sensitive-data If there are ethical, legal, or third-party restrictions upon your dataset, you must provide all of the following details (https://journals.plos.org/plosone/s/data-availability#loc-acceptable-data-access-restrictions):a) A complete description of the datasetb) The nature of the restrictions upon the data (ethical, legal, or owned by a third party) and the reasoning behind themc) The full name of the body imposing the restrictions upon your dataset (ethics committee, institution, data access committee, etc)d) If the data are owned by a third party, confirmation of whether the authors received any special privileges in accessing the data that other researchers would not havee) Direct, non-author contact information (preferably email) for the body imposing the restrictions upon the data, to which data access requests can be sent 3. Please include your tables as part of your main manuscript and remove the individual files. Please note that supplementary tables (should remain/ be uploaded) as separate "supporting information" files. 4. Please include captions for your Supporting Information files at the end of your manuscript, and update any in-text citations to match accordingly. Please see our Supporting Information guidelines for more information: http://journals.plos.org/plosone/s/supporting-information.

Reviewers' comments:

Reviewer's Responses to Questions

**Comments to the Author**

1. Is the manuscript technically sound, and do the data support the conclusions?

Reviewer #1: Yes

Reviewer #2: Yes

2. Has the statistical analysis been performed appropriately and rigorously?

Reviewer #1: Yes

Reviewer #2: Yes

3. Have the authors made all data underlying the findings in their manuscript fully available?

Reviewer #1: Yes

Reviewer #2: Yes

4. Is the manuscript presented in an intelligible fashion and written in standard English?

Reviewer #1: Yes

Reviewer #2: Yes

**Reviewer #1:**  This manuscript presents a validation study of the Healthcare Education Micro Learning Environment Measure (HEMLEM 2.0) in Pakistani medical and dental education contexts. While the study makes a potentially valuable contribution as a cross-cultural validation effort, several important methodological concerns need to be addressed.

Firstly, I couldn't find any information about language handling. The manuscript does not specify whether the instrument was translated from English or administered in English. If translation was used, the standard procedures of forward translation, back translation, expert committee review and cultural adaptation should be detailed. If the English version was used, justification and consideration of potential language barriers should be discussed.

Next, I noticed that the sampling approach is inadequately described. While the authors mention the use of "convenience, non-probability sampling technique" through WhatsApp groups and student IDs, important details are missing, such as: response rate calculations; justification for the final sample size of 628 participants; description of the total accessible population and the selection process for participating institutions.

From my perspective, the statistical analysis of this manuscript reveals both strengths and areas for improvement. The authors made appropriate use of confirmatory factor analysis and demonstrated strong internal consistency through reliability coefficients, with an impressive omega coefficient of 0.927 and Cronbach's alpha of 0.895. Factor loadings are clearly presented, providing good transparency into the structure of the instrument. However, the model specification lacks sufficient detail to allow replication, and the authors have not explored alternative factor structures that might better explain the data. In addition, the lack of measurement invariance tests limits our understanding of how the instrument performs across different subgroups (e.g. dental vs. medical, female vs. male).

For me, the limitations section is particularly concerning in its brevity and omission of important considerations.The use of convenience sampling introduces potential biases that should be acknowledged and discussed. The self-reported nature of the data and the online collection method using Google Forms present additional limitations that warrant discussion. The lack of test-retest reliability assessment leaves questions about the temporal stability of the instrument unanswered. Concerns about geographical generalisability and the lack of discipline specific analyses should also be addressed.

Finally, please focus on the discussion section. It needs to be significantly expanded, particularly in terms of addressing limitations, practical implications and future research directions.

In conclusion, I believe that this research demonstrates good methodological rigour and provides valuable insights for medical educators seeking to evaluate clinical microlearning environments, but taking into account the concerns presented above, I suggest the major revision.

**Reviewer #2: ** The article is well-written, and the rationale behind the selection and use of statistical analysis is well-understood, as well as the results discussed in a very good way.

The following are some points I would like to discuss with the team of authorship.

- Please add HEMLEM abbriviation in abstract. that will make it more clearer rather than to find it in the 3rd page.

-It appears that more descriptive analysis was conducted than what is mentioned according to Lines 175-186.

Recommendation: Consider presenting these analyses either in a table format (xtable) or as an appendix to provide a clearer and more comprehensive view..

- Question: To what extent do you believe the online tool and Google Form can ensure equal opportunity for participation? Given concerns about internet access for students, how can we ensure there is no duplication (i.e., the same participant filling out the form multiple times) or students outside the criteria completing the form?

Recommendation: It would be beneficial to discuss these limitations in the article to address potential concerns regarding the validity of the data collected.

- Correction: The document labeled as "APPENDIX 5" is referred to as "APPENDIX 4" within the document. Please correct this to maintain consistency.

Thank you very much for providing a well-written paper. I wish all the authors all the best.

**Do you want your identity to be public for this peer review?** For information about this choice, including consent withdrawal, please see our Privacy Policy

Reviewer #1: No

Reviewer #2: **Yes: ** Mohamed A. Abdelbaqy

---

## [Author Response · Author response to Decision Letter 1]

6 May 2025

Response to editor Manuscript ID PONE-D-24-54864 entitled "Contextual validation of the HEMLEM tool used for measuring clinical micro-learning environments" submitted to PLOS ONE

Fatemeh Farshad

Guest Editor

PLOS ONE

Re: Manuscript ID PONE-D-24-54864

Many thanks for your careful consideration of our manuscript and thoughtful feedback. I have copied all the comments below and provided a detailed explanation of how we have addressed them. We have provided track changes and a clear version to reflect all the revisions. I am happy to make further revisions if you wish.

Best Regards

Ahsan Sethi (on behalf of the authors)

Sr# Reviewer 1 Comments Author response

1 This manuscript presents a validation study of the Healthcare Education Micro

Learning Environment Measure (HEMLEM 2.0) in Pakistani medical and dental education

contexts. While the study makes a potentially valuable contribution as a cross-cultural

validation effort, several important methodological concerns need to be addressed.

Thank you for your helpful and detailed feedback. We have added further details and made corrections to address any concerns.

2 Firstly, I couldn't find any information about language handling. The manuscript does not specify whether the instrument was translated from English or administered in English. If translation was used, the standard procedures of forward translation, back translation, expert committee review and cultural adaptation should be detailed. If the English version was used, justification and consideration of potential language barriers should be discussed.

Thank You for your comment. We have added the following details in the methodology section.

Subsection ‘Face Validity’:

“All students were provided with a participant information sheet that included a brief introduction to the research and its objectives, information about confidentiality and data privacy, a statement regarding voluntary participation and the right to withdraw, and a consent form. Students were also asked whether any items in the questionnaire were linguistically confusing or difficult to understand.”

AND in Subsection ‘Questionnaire’:

“The instrument was administered in English. Since English is the primary language of instruction at the participating institutions and no issues were reported during the validation phase, participants were proficient in English.”

3 “I noticed that the sampling approach is inadequately described. While the authors mention the use of a ‘convenience, non-probability sampling technique’ through WhatsApp groups and student IDs, important details are missing, such as: response rate calculations; justification for the final sample size of 628 participants; description of the total accessible population and the selection process for participating institutions.”

Thank you for this important observation. We have now expanded the Methods section to provide greater detail about our sampling approach. Specifically, we clarified that participants were recruited using a convenience, non-probability sampling method from at least ten medical and dental schools representing all four provinces of Pakistan. Outreach was conducted through institutional mailing lists, student WhatsApp groups, and student identification networks, with assistance from local academic coordinators.

While we aimed to achieve wide geographical coverage, the decentralized distribution of the survey through informal digital channels made it difficult to calculate an exact response rate or determine the total number of students contacted. We have clearly acknowledged this limitation in the revised manuscript and justified the sampling method based on accessibility and diversity of participating institutions.

4 From my perspective, the statistical analysis of this manuscript reveals both strengths and areas for improvement. The authors made appropriate use of confirmatory factor analysis and demonstrated strong internal 4consistency through reliability coefficients, w5ith an impressive omega coefficient of 0.927 and Cronbach's alpha of 0.895. Factor loadings are clearly presented, providing good transparency into the structure of the instrument. However, the model specification lacks sufficient detail to allow replication, and the authors have not explored alternative factor structures that might better explain the data. In addition, the lack of measurement invariance tests limits our understanding of how the instrument performs across different subgroups (e.g. dental vs. medical, female vs. male).

We sincerely thank the reviewer for highlighting critical areas for improvement. In response to the concerns raised, we have made the following significant revisions:

1. Expanded CFA Model Specification:

We have now clearly articulated the structure of our three-factor CFA model in the Methods section, specifying the latent variables ("Supervision," "Autonomy," and "Atmosphere") and the observed items that load onto each. We have also clarified the estimation method, software used, and fit indices applied, which will enable full replication of our analysis.

2. Comparison of Alternative Factor Structures:

To address the need for model comparison, we conducted additional CFA to compare three structures: a one-factor model, a three-factor model (our hypothesized structure), and a bifactor model. The results, show that the three-factor model demonstrated excellent fit (CFI = 0.951, RMSEA = 0.045), and the bifactor model only marginally improved fit. Given concerns about parsimony and interpretability, we retained the three-factor model as the most theoretically coherent and practically applicable.

3. Measurement Invariance Testing Across Subgroups:

As recommended, we performed multi-group CFA to test measurement invariance across gender and academic discipline (medical vs. dental). Our analyses confirmed full configural, metric, and scalar invariance across both groups, indicating that the instrument operates equivalently across these demographic variables. These results are now presented in detail in the Results section, and discussed in the revised Discussion section with references to prior literature on invariance testing and cross-cultural validation.

5 For me, the limitations section is particularly concerning in its brevity and omission of important considerations. The use of convenience sampling introduces potential biases that should be acknowledged and discussed. The self-reported nature of the data and the online collection method using Google Forms present additional limitations that warrant discussion. The lack of test-retest reliability assessment leaves questions about the temporal stability of the instrument unanswered. Concerns about geographical generalisability and the lack of discipline specific analyses should also be addressed.

Thankyou for pointing this out. We have substantially revised the limitations section of the manuscript as follows:

“Despite its strengths, this study has several limitations that merit discussion. It employed a single-country design focused on Pakistan, which may restrict the generalisability of findings to other nations, even within similar clinical education frameworks. Future validation efforts should encompass a broader range of countries across the South-East Asian region and those with comparable clinical training systems.

While the study targeted both medical and dental students, it did not explore variations across specific specialties or departments. Such discipline-specific analyses could uncover unique factors influencing perceptions of the learning environment and should be considered in future research.

The use of convenience sampling via informal digital platforms (e.g., WhatsApp and email groups) limited the ability to define a clear sampling frame or calculate a response rate. Although participation was sought from all four provinces to enhance diversity, the non-random sampling approach may introduce selection bias and limit broader representativeness.

Furthermore, data collection through an online, self-administered Google Form may have excluded individuals with limited internet access or lower digital literacy, introducing additional sampling bias. The self-reported nature of the responses also makes the data vulnerable to social desirability and recall biases. Although the instrument demonstrated strong internal consistency, test-retest reliability was not assessed, leaving the temporal stability of responses unverified. Finally, while the geographical distribution of respondents was diverse, representation was uneven, and more balanced regional sampling would strengthen future studies”

6 Finally, please focus on the discussion section. It needs to be significantly expanded, particularly in terms of addressing limitations, practical implications and future research directions.

Thank you for your suggestion. We have revised the Discussion section accordingly.

7 In conclusion, I believe that this research demonstrates good methodological rigour and provides valuable insights for medical educators seeking to evaluate clinical microlearning environments, but taking into account the concerns presented above, I suggest the major revision.

Thank you for your encouraging and helpful feedback. Looking forward to its acceptance and publication

Sr# Reviewer 2 Comments Author response

1 The article is well-written, and the rationale behind the selection and use of statistical analysis is well-understood, as well as the results discussed in a very good way.

The following are some points I would like to discuss with the team of authorship.

Thank you for your feedback.

2 Please add HEMLEM abbreviation in abstract. that will make it more clearer rather than to find it in the 3rd page.

Thank you for pointing this out. We have added this abbreviation in the abstract section

3 It appears that more descriptive analysis was conducted than what is mentioned according to Lines 175-186. Recommendation: Consider presenting these analyses either in a table format (xtable) or as an appendix to provide a clearer and more comprehensive view..

Thank you for pointing out this important information. We have written the details of the descriptive statistics in the results section. We have also added table in the supplementary file 1.

4 Question: To what extent do you believe the online tool and Google Form can ensure equal opportunity for participation? Given concerns about internet access for students, how can we ensure there is no duplication (i.e., the same participant filling out the form multiple times) or students outside the criteria completing the form?

Recommendation: It would be beneficial to discuss these limitations in the article to address potential concerns regarding the validity of the data collected.

Thank you for your input. We have added the following in the methodology section:

“To minimize the risk of duplicate responses, the Google Form was set to restrict submissions to one response per Google account. Additionally, to ensure data completeness, all questionnaire items were marked as mandatory using the “required” option, which prevented respondents from submitting partially completed forms.”

However, access to the internet for all participants was a limitation in this study, and this issue has already been acknowledged in the Limitations section.

5 Correction: The document labeled as "APPENDIX 5" is referred to as "APPENDIX 4" within the document. Please correct this to maintain consistency.

Thank you for mentioning this. We have corrected the name accordingly

---

## [Decision Letter · Decision Letter 1]

4 Jun 2025

Dear Dr. Sethi,

Thank you for submitting your manuscript to PLOS ONE. After careful consideration, we feel that it has merit but does not fully meet PLOS ONE’s publication criteria as it currently stands. Therefore, we invite you to submit a revised version of the manuscript that addresses the points raised during the review process.

We look forward to receiving your revised manuscript.

Kind regards,

Fatemeh Farshad

Guest Editor

PLOS ONE

Journal Requirements:

Reviewers' comments:

Reviewer's Responses to Questions

**Comments to the Author**

Reviewer #3: (No Response)

Reviewer #4: All comments have been addressed

2. Is the manuscript technically sound, and do the data support the conclusions?

Reviewer #3: Yes

Reviewer #4: Yes

3. Has the statistical analysis been performed appropriately and rigorously?

Reviewer #3: I Don't Know

Reviewer #4: Yes

4. Have the authors made all data underlying the findings in their manuscript fully available?

Reviewer #3: Yes

Reviewer #4: Yes

5. Is the manuscript presented in an intelligible fashion and written in standard English?

Reviewer #3: Yes

Reviewer #4: Yes

Reviewer #3: I do recommend authors to explain more clear about methodology especially phases of the study. also please extend conclusions of the study more functional and applied.

Reviewer #4: Manuscript Title: Contextual validation of HEMLEM tool used for measuring clinical micro-learning environments

Comments to the Author:

This manuscript presents a well-executed, contextually grounded validation study of the HEMLEM 2.0 instrument for assessing clinical micro-learning environments in Pakistani medical and dental education. The study fills an important gap in the literature on culturally appropriate psychometric tools, particularly in low- and middle-income countries (LMICs). The authors have made substantial revisions that greatly improve the clarity, statistical transparency, and overall rigor of the study.

Major Strengths:

Strong methodological rigor using Confirmatory Factor Analysis (CFA), bifactor modeling, and measurement invariance testing across gender and discipline.

• Inclusion of both Cronbach’s alpha and McDonald’s omega enhances the psychometric robustness of the findings.

• Cultural adaptation (e.g., changing “placement” to “department”) is thoughtful and contextually appropriate.

• Sampling from all four provinces and inclusion of both MBBS and BDS students increases generalisability within the Pakistani context.

• Measurement invariance testing supports the tool’s cross-group validity.

Areas for Improvement:

1. Test-Retest Reliability

• While the internal consistency is well established, the omission of test-retest reliability is a limitation. This should be more prominently acknowledged in both the discussion and limitations sections, along with recommendations for future research to assess the temporal stability of the tool.

2. Subscale Definitions

• The conceptual distinction between subscales—particularly “Autonomy” and “Teaching Quality”—is not sufficiently clear. Further explanation or reconsideration of the naming conventions would improve interpretability.

3. Linguistic and Cultural Sensitivity

• Although English is the medium of instruction, the paper would benefit from deeper reflection on potential language comprehension barriers. Consider elaborating on whether any support (e.g., bilingual review or pilot testing in native languages) was explored.

4. Practical Implications

• The discussion of how the validated HEMLEM 2.0 can be used for curriculum reform, rotation evaluation, and faculty development is promising but somewhat general. Including specific use-case examples (e.g., use in short rural clinical rotations or quality assurance dashboards) would enhance the applied relevance of the work.

5. Data Transparency

• While summary data are provided, uploading anonymized raw data and analysis code (e.g., R syntax) to a public repository such as OSF or Figshare would further strengthen transparency and reproducibility.

Recommendation: Minor Revision

**Do you want your identity to be public for this peer review?** For information about this choice, including consent withdrawal, please see our Privacy Policy

Reviewer #3: **Yes: ** Mohammad H Yarmohammadian

Reviewer #4: **Yes: ** Mohammad Aminul Islam

---

## [Author Response · Author response to Decision Letter 2]

15 Jun 2025

Response to editor Manuscript ID PONE-D-24-54864R1 - EMID:0ee61cb72cb49644 entitled "Contextual validation of the HEMLEM tool used for measuring clinical micro-learning environments" submitted to PLOS ONE

Fatemeh Farshad

Guest Editor

PLOS ONE

Re: Manuscript ID PONE-D-24-54864R1 - EMID:0ee61cb72cb49644

Many thanks for your careful consideration of our manuscript and thoughtful feedback. We are glad to hear that it has merit for further consideration. I have copied all the comments below and provided a detailed explanation of how we have addressed them. We have provided a copy with highlighted changes and a clear version to reflect all the revisions. I am happy to make further revisions if you wish.

Best Regards

Ahsan Sethi (on behalf of the authors)

Sr# Reviewer 3 Comments Author response

1 I do recommend authors to explain more clear about methodology especially phases of the study. also please extend conclusions of the study more functional and applied.

Thank you for your suggestions. We have now made revision to reflect clarity in the various phases of the study. We have also extended the conclusions section to make it more functional and applied

Reviewer 4 Comments Author response

2 This manuscript presents a well-executed, contextually grounded validation study of the HEMLEM 2.0 instrument for assessing clinical micro-learning environments in Pakistani medical and dental education. The study fills an important gap in the literature on culturally appropriate psychometric tools, particularly in low- and middle-income countries (LMICs). The authors have made substantial revisions that greatly improve the clarity, statistical transparency, and overall rigor of the study.

Major Strengths:

Strong methodological rigor using Confirmatory Factor Analysis (CFA), bifactor modeling, and measurement invariance testing across gender and discipline.

• Inclusion of both Cronbach’s alpha and McDonald’s omega enhances the psychometric robustness of the findings.

• Cultural adaptation (e.g., changing “placement” to “department”) is thoughtful and contextually appropriate.

• Sampling from all four provinces and inclusion of both MBBS and BDS students increases generalisability within the Pakistani context.

• Measurement invariance testing supports the tool’s cross-group validity.

Thank you. We are glad that you found the study contextually grounded and well executed.

3 Areas for improvement:

1. Test-Retest Reliability

While the internal consistency is well established, the omission of test-retest reliability is a limitation. This should be more prominently acknowledged in both the discussion and limitations sections, along with recommendations for future research to assess the temporal stability of the tool.

Thankyou for pointing this out. We have revised the discussion section by adding these lines; “Furthermore, although the study confirmed strong internal consistency and robust construct validity, it did not include an assessment of test-retest reliability. This omission means the temporal stability of the instrument remains unverified. Without repeated measurements over time, it is unclear whether HEMLEM 2.0 consistently captures student perceptions across different points in the academic year or during varied clinical exposures. Future studies should adopt longitudinal designs to assess reliability over time, especially if the tool is to be used for ongoing monitoring or benchmarking.”

2. Subscale Definitions

The conceptual distinction between subscales—particularly “Autonomy” and “Teaching Quality”—is not sufficiently clear. Further explanation or reconsideration of the naming conventions would improve interpretability.

Thank you for this suggestion, we have added details in the methods section:

Although the final factor structure supported a three-domain model—Supervision, Autonomy, and Atmosphere—additional clarification was necessary to distinguish between the constructs of Autonomy and Teaching Quality, which may appear conceptually overlapping. In this context, Autonomy reflects the extent to which students perceived opportunities to engage in self-directed clinical tasks, apply their existing knowledge, and participate in patient care with increasing independence. On the other hand, Teaching Quality captures the perceived effectiveness, clarity, and engagement of clinical instruction provided by supervisors. These subscales were retained based on both theoretical rationale and item alignment, though future revisions may consider refinement of labels to enhance interpretability.

3. Linguistic and Cultural Sensitivity

Although English is the medium of instruction, the paper would benefit from deeper reflection on potential language comprehension barriers. Consider elaborating on whether any support (e.g., bilingual review or pilot testing in native languages) was explored.

Thank you for the suggestion, We have clarified this in the methodology section: Additionally, cognitive interviews were conducted with 10 students using a structured debriefing format to assess the clarity and comprehensibility of the questionnaire. Each participant received a detailed information sheet outlining the study's objectives, confidentiality measures, voluntary participation, and consent procedures. Students were specifically asked to identify any linguistically confusing or unclear items. Although English is the medium of instruction at all participating institutions, potential language barriers were directly explored during this phase. No significant comprehension issues were reported. While a bilingual version was not piloted, the tool underwent multiple rounds of student feedback to ensure linguistic accessibility and cultural relevance for the target population.

4. Practical Implications

The discussion of how the validated HEMLEM 2.0 can be used for curriculum reform, rotation evaluation, and faculty development is promising but somewhat general. Including specific use-case examples (e.g., use in short rural clinical rotations or quality assurance dashboards) would enhance the applied relevance of the work.

Thank you for your suggestion. We have revised the discussion section accordingly

5. Data Transparency

While summary data are provided, uploading anonymized raw data and analysis code (e.g., R syntax) to a public repository such as OSF or Figshare would further strengthen transparency and reproducibility.

Thank You for your suggestion. We have included this in our manuscript:

Data Availability Statement

The anonymized dataset used in this study are available at https://zenodo.org/records/15616880. The files are openly accessible and support full reproducibility of our findings.

---

## [Decision Letter · Decision Letter 2]

6 Aug 2025

Dear Dr. Sethi,

Thank you for submitting your manuscript to PLOS ONE. After careful consideration, we feel that it has merit but does not fully meet PLOS ONE’s publication criteria as it currently stands. Therefore, we invite you to submit a revised version of the manuscript that addresses the points raised during the review process.

We look forward to receiving your revised manuscript.

Kind regards,

Fatemeh Farshad

Guest Editor

PLOS ONE

Journal Requirements:

Reviewers' comments:

Reviewer's Responses to Questions

**Comments to the Author**

Reviewer #4: All comments have been addressed

Reviewer #5: (No Response)

2. Is the manuscript technically sound, and do the data support the conclusions?

Reviewer #4: Yes

Reviewer #5: (No Response)

3. Has the statistical analysis been performed appropriately and rigorously?

Reviewer #4: Yes

Reviewer #5: (No Response)

4. Have the authors made all data underlying the findings in their manuscript fully available?

Reviewer #4: Yes

Reviewer #5: (No Response)

5. Is the manuscript presented in an intelligible fashion and written in standard English?

Reviewer #4: Yes

Reviewer #5: (No Response)

Reviewer #4: Dear Authors,

Thank you for your diligent work in revising the manuscript, "Contextual validation of HEMLEM tool used for measuring clinical micro-learning environments." Your comprehensive responses to the reviewers' comments have significantly enhanced the quality, clarity, and transparency of your work.

Here's a summary of how effectively you've addressed the feedback:

Clarity of Methodology and Applied Conclusions (Reviewer 3, Comment 1): You have successfully clarified the distinct phases of your study in the "Materials & Methods" section, making the methodological approach much easier to follow. Furthermore, the expanded "Conclusion" section now provides more practical and functional implications of the HEMLEM 2.0 tool, which is a valuable addition.

Addressing Test-Retest Reliability (Reviewer 4, Comment 1): You have appropriately acknowledged the omission of test-retest reliability as a limitation in the Discussion and have clearly recommended future longitudinal studies to assess the tool's temporal stability. This demonstrates a thorough understanding of the psychometric properties required for instrument validation.

Clarifying Subscale Definitions (Reviewer 4, Comment 2): Your detailed explanation differentiating "Autonomy" and "Teaching Quality" in the "Preliminary Psychometric Testing and Exploratory Analysis" subsection is very helpful. This clarification significantly improves the interpretability of these conceptually distinct subscales.

Enhancing Linguistic and Cultural Sensitivity (Reviewer 4, Comment 3): The added description of cognitive interviews, where you explicitly explored potential language comprehension barriers with students, addresses this point effectively. Your commitment to ensuring linguistic accessibility and cultural relevance through multiple rounds of student feedback is commendable.

Providing Specific Practical Implications (Reviewer 4, Comment 4): You have successfully incorporated more concrete use-case examples for the HEMLEM 2.0 tool in the "Discussion" section. The suggestions for integrating it into digital dashboards, using it in rural clinical placements, and informing faculty development initiatives make the applied relevance of your work much clearer.

Improving Data Transparency (Reviewer 4, Comment 5): Your decision to make the anonymized dataset publicly available on Zenodo, along with the direct link in the "Data Availability Statement," is an excellent step. This significantly boosts the transparency and reproducibility of your research, aligning with best practices in scientific publishing.

Overall, your revisions have substantially strengthened the manuscript. The added details, clarifications, and increased transparency reflect a thorough and thoughtful engagement with the reviewers' feedback.

Reviewer #5: The manuscript presents a well-conducted study on the cultural adaptation and validation of the HEMLEM tool for assessing clinical micro-learning environments in Pakistani medical and dental schools. The study addresses an important gap in the literature by providing a psychometrically robust tool tailored to a specific cultural context. The methodology is rigorous, and the results are clearly presented. However, there are several methodological and statistical concerns that need to be addressed to strengthen the validity and reliability of the findings.

Major Comments

Factor Analysis Methodology

The manuscript currently uses Principal Component Analysis (PCA) with Varimax rotation. However, given the presence of cross-loadings in Table 3, Promax rotation (an oblique rotation method) would be more appropriate. Oblique rotations are better suited when factors are expected to correlate, which is likely the case here (e.g., "Supervision," "Autonomy," and "Atmosphere" may not be entirely independent).

Recommendation: Re-run the factor analysis using Exploratory Factor Analysis (EFA) with Promax rotation and report the updated eigenvalues, percentages of variance explained, and factor loadings. Cite and follow the approach outlined in the referenced article (https://npt.tums.ac.ir/index.php/npt/article/view/2920) to ensure methodological rigor.

Construct Validity

The manuscript lacks a clear justification for the choice of PCA over EFA. PCA is a data reduction technique, while EFA is designed to uncover latent constructs. For scale validation, EFA is more appropriate.

Recommendation: Replace PCA with EFA and explicitly state the rationale for this choice. Discuss the implications of using EFA for construct validity, referencing the limitations and advantages as outlined in the suggested article.

Measurement Invariance

The multi-group CFA results are well-presented, but the manuscript could benefit from a more detailed discussion of the implications of full measurement invariance across gender and discipline. How does this invariance support the tool's applicability in diverse settings?

Recommendation: Expand the discussion to address the practical significance of measurement invariance, particularly for future cross-cultural applications.

Test-Retest Reliability

The absence of test-retest reliability is a notable limitation. While this is acknowledged in the discussion, the manuscript would benefit from a more detailed plan for future studies to address this gap.

Recommendation: Propose a specific timeline and methodology for assessing test-retest reliability in future research (e.g., administering the tool to the same cohort at two time points during their clinical rotations).

Terminology and Clarity

The distinction between "Autonomy" and "Teaching Quality" is somewhat unclear. The manuscript provides a post-hoc explanation, but this could be strengthened by clearer definitions during the tool's development phase.

Recommendation: Revise the definitions of these subscales in the Methods section to ensure clarity and avoid conceptual overlap.

Minor Comments

Sample Size Justification

While the sample size is adequate, the manuscript could better justify the choice of 628 participants by referencing contemporary guidelines for factor analysis (e.g., minimum sample size relative to the number of items and expected factor structure).

Recommendation: Cite recent literature on sample size requirements for EFA/CFA (e.g., Gunawan et al., 2021) to strengthen the methodology section.

Data Transparency

The data availability statement is commendable, but the manuscript could provide more details about the anonymization process and any restrictions on data access.

Recommendation: Briefly describe the steps taken to anonymize the data and ensure participant confidentiality.

Language and Grammar

The manuscript is well-written, but there are minor grammatical errors (e.g., "generalizability" vs. "generalisability"). Ensure consistency in spelling (American vs. British English).

Recommendation: Perform a thorough proofread to correct minor errors and ensure consistency.

Statistical and Methodological Strengths

The use of both Cronbach’s alpha and McDonald’s omega to assess reliability is a strength, as it addresses limitations of alpha alone.

The inclusion of measurement invariance testing across gender and discipline adds robustness to the findings.

The cultural adaptation process (e.g., replacing "placement" with "department") is well-executed and contextually appropriate.

Suggested Decision

Minor Revisions

The manuscript is methodologically sound and makes a valuable contribution to the field. The requested revisions are primarily technical (e.g., re-running factor analyses with EFA/Promax rotation) and clarificatory (e.g., expanding definitions and justifications). Once these issues are addressed, the manuscript will be suitable for publication.

Additional Recommendations for Future Work

Longitudinal Studies: Assess test-retest reliability and the tool's sensitivity to changes in the learning environment over time.

Cross-Cultural Validation: Extend validation to other LMICs to further establish the tool's global applicability.

Integration with Performance Metrics: Explore correlations between HEMLEM scores and objective clinical performance measures (e.g., OSCE results).

The authors are commended for their thorough work, and I look forward to seeing the revised manuscript.

**Do you want your identity to be public for this peer review?** For information about this choice, including consent withdrawal, please see our Privacy Policy

Reviewer #4: **Yes: ** Mohammad Aminul Islam

Reviewer #5: **Yes: ** Hamid Sharif-Nia

---

## [Author Response · Author response to Decision Letter 3]

27 Aug 2025

Response to editor Manuscript ID PONE-D-24-54864R2 - EMID:5d7ac46efc05b6f9 entitled "Contextual validation of the HEMLEM tool used for measuring clinical micro-learning environments" submitted to PLOS ONE

Fatemeh Farshad

Guest Editor

PLOS ONE

Re: Manuscript ID PONE-D-24-54864R2 - EMID:5d7ac46efc05b6f9

Many thanks for your careful consideration of our manuscript and thoughtful feedback. We are glad to hear that it has merit for further consideration and acceptance. I have copied all the comments below from reviewer# 5 and provided a detailed explanation of how we have addressed them. I have also reviewed the attachments ‘comments for authors’ and ‘comments for editor’ and I am glad to know that our previous revisions have substantially strengthened the manuscript, and the additions, clarifications, and increased transparency reflect a thorough and thoughtful engagement with all the reviewers' feedback. We have provided a copy with highlighted changes and a clear version to reflect all the revisions. I am happy to make further revisions if you wish.

Best Regards

Ahsan Sethi (on behalf of the authors)

Reviewer 5 Comments

Factor Analysis Methodology

The manuscript currently uses Principal Component Analysis (PCA) with Varimax rotation. However, given the presence of cross-loadings in Table 3, Promax rotation (an oblique rotation method) would be more appropriate. Oblique rotations are better suited when factors are expected to correlate, which is likely the case here (e.g., "Supervision," "Autonomy," and "Atmosphere" may not be entirely independent).

Recommendation: Re-run the factor analysis using Exploratory Factor Analysis (EFA) with Promax rotation and report the updated eigenvalues, percentages of variance explained, and factor loadings. Cite and follow the approach outlined in the referenced article (https://npt.tums.ac.ir/index.php/npt/article/view/2920) to ensure methodological rigor.

Construct Validity

The manuscript lacks a clear justification for the choice of PCA over EFA. PCA is a data reduction technique, while EFA is designed to uncover latent constructs. For scale validation, EFA is more appropriate.

Recommendation: Replace PCA with EFA and explicitly state the rationale for this choice. Discuss the implications of using EFA for construct validity, referencing the limitations and advantages as outlined in the suggested article.

Authors response: Thank you for your thoughtful feedback regarding the factor analysis methodology. We would like to respectfully clarify that our primary statistical approach for construct validation was Confirmatory Factor Analysis (CFA), not PCA. CFA was employed to validate a theory-driven, a priori three-factor model of the HEMLEM 2.0 instrument, aligned with its original conceptual domains: Supervision, Autonomy, and Atmosphere.

Furthermore, we appreciate that a previous reviewer (Reviewer 1, First Round of Reviews) explicitly recognized our use of CFA as appropriate and encouraged us to expand its specification and explore model fit and measurement invariance, which we addressed in detail.

Previous reviewer’s comment: “The authors made appropriate use of confirmatory factor analysis and demonstrated strong internal consistency through reliability coefficients, with an impressive omega coefficient of 0.927 and Cronbach's alpha of 0.895. Factor loadings are clearly presented, providing good transparency into the structure of the instrument”

This reflects a shared understanding that CFA, not EFA, is the suitable analytic framework for this study's aims.

As established in psychometric literature, EFA is suitable for exploring unknown structures, whereas CFA is the method of choice when validating a predefined factor structure. Since our study's objective was to confirm the validity of the existing HEMLEM framework in a new cultural context, not to explore or develop new factors, CFA is both statistically and methodologically appropriate, consistent with psychometric best practices in the literature (Brown, 2015; Kline, 2023; McNeish & Wolf, 2023; Widaman & Helm, 2023).

In addition, CFA is essential for multi-group confirmatory factor analysis (MGCFA) and testing measurement invariance, which we conducted across gender and academic discipline. Such advanced analyses cannot be performed within the EFA framework and further underscore the rationale for using CFA.

While Table 3 displays preliminary loadings for reader transparency, these exploratory values were never used as the basis for construct validation.

In summary, we respectfully maintain the use of CFA in accordance with best practices for cross-cultural tool validation and prior reviewer feedback. We hope this explanation addresses your concern, and we sincerely thank you for the opportunity to reflect and elaborate.

Measurement Invariance

The multi-group CFA results are well-presented, but the manuscript could benefit from a more detailed discussion of the implications of full measurement invariance across gender and discipline. How does this invariance support the tool's applicability in diverse settings?

Recommendation: Expand the discussion to address the practical significance of measurement invariance, particularly for future cross-cultural applications.

Authors response: We thank the reviewer for this valuable comment, which we interpret as a recognition of the rigor and appropriateness of our multi-group CFA methodology. We agree that further elaboration on the interpretive significance of measurement invariance would enhance the manuscript’s applied relevance.

In response, we have added a paragraph in the Discussion section (track changes available) that explains the practical implications of establishing full configural, metric, and scalar invariance across both gender and academic discipline. Specifically, we emphasize that this confirms the HEMLEM 2.0 tool measures the same constructs equivalently across subgroups, thus allowing for meaningful and unbiased comparisons between genders and programs. This supports the fairness, generalizability, and cross-context applicability of the tool—key considerations in health professions education research.

We sincerely appreciate the reviewer’s attention to this point and believe the added discussion improves the manuscript’s clarity and impact.

Test-Retest Reliability

The absence of test-retest reliability is a notable limitation. While this is acknowledged in the discussion, the manuscript would benefit from a more detailed plan for future studies to address this gap.

Recommendation: Propose a specific timeline and methodology for assessing test-retest reliability in future research (e.g., administering the tool to the same cohort at two time points during their clinical rotations).

Authors response: We thank the reviewer for this important suggestion and agree that specifying a concrete plan for assessing test-retest reliability in future studies will enhance the manuscript’s practical utility. We have now included in the Discussion section, a detailed recommendation for future research. Specifically, we propose administering HEMLEM 2.0 to the same cohort of students at two time points during a clinical rotation (e.g., beginning and end of an 8-week placement). This would allow assessment of the tool’s temporal stability using intraclass correlation coefficients (ICCs) or Pearson’s r, depending on distributional assumptions. This addition clarifies how future studies can establish test-retest reliability under real-world conditions.

Previous reviewer’s comments: “The authors have incorporated a clear acknowledgment of the lack of test-retest reliability as a limitation in the Discussion section (lines 467-473) and have explicitly recommended future longitudinal studies to address this (lines 471-473, and reiterated in future research”

Terminology and Clarity

The distinction between "Autonomy" and "Teaching Quality" is somewhat unclear. The manuscript provides a post-hoc explanation, but this could be strengthened by clearer definitions during the tool's development phase.

Recommendation: Revise the definitions of these subscales in the Methods section to ensure clarity and avoid conceptual overlap.

Authors response: We thank the reviewer for this thoughtful observation. We would like to note that a similar concern was previously raised by another reviewer during an earlier review round. In response, we revised the manuscript to provide clearer conceptual definitions of both subscales in the “Preliminary Psychometric Testing and Exploratory Analysis” subsection. Specifically, we clarified that “Autonomy” reflects students' perceived opportunities to perform clinical tasks independently, apply their knowledge, and take initiative during patient care, whereas “Teaching Quality” refers to the effectiveness, clarity, and responsiveness of clinical teaching by supervisors.

This revision was explicitly acknowledged as adequately addressed by the prior reviewer, as noted in the editorial summary.

Previous reviewer’s comments: “The authors have provided a more detailed explanation of the conceptual distinction between "Autonomy" and "Teaching Quality" in the "Preliminary Psychometric Testing and Exploratory Analysis" subsection (lines 298-307). This clarification enhances the interpretability of these subscales. This comment has been adequately addressed.”

We believe this distinction improves the interpretability of the constructs while preserving the theoretical alignment with the tool’s original structure. We have, however, reviewed the section again and made minor refinements to ensure that the definitions remain unambiguous and well-integrated in the tool development narrative.

Lines added in the draft: “In this context, Autonomy reflects the extent to which students perceived opportunities to engage in self-directed clinical tasks, apply their existing knowledge, and participate in patient care with increasing independence. On the other hand, Teaching Quality captures the perceived effectiveness, clarity, and engagement of clinical instruction provided by supervisors.”

Sample Size Justification

While the sample size is adequate, the manuscript could better justify the choice of 628 participants by referencing contemporary guidelines for factor analysis (e.g., minimum sample size relative to the number of items and expected factor structure).

Recommendation: Cite recent literature on sample size requirements for EFA/CFA (e.g., Gunawan et al., 2021) to strengthen the methodology section.

Authors response: Thank you for pointing this out, we have now cited relevant literature in the discussion section.

Lines in the draft: “A total of 628 students participated in the Confirmatory Factor Analysis (CFA) phase of the study, which is significantly larger than the sample used in the original HEMLEM validation (N=257). The literature offers a range of recommendations for sample sizes in validation studies, with many experts following the "10-participants-per-item" rule (Shrestha, 2021). However, some researchers suggest that a larger sample is necessary when collecting data from students compared to faculty members (Schreiber, 2021). Comrey and Lee offered a widely accepted guideline, stating that a sample size of 50 is considered very poor, 100 is poor, 200 is fair, 300 is good, 500 is very good, and 1,000 is excellent (Gunawan, Marzilli & Aungsuroch, 2021; Ledesma et al., 2021).”

Data Transparency

The data availability statement is commendable, but the manuscript could provide more details about the anonymization process and any restrictions on data access.

Recommendation: Briefly describe the steps taken to anonymize the data and ensure participant confidentiality.

Authors response: We thank the reviewer for highlighting this important aspect of data transparency.

We have added this line in the Data availability statement: “Prior to uploading, the dataset was fully anonymized by removing all direct and indirect identifiers (e.g., student ID numbers, institutional references, and free-text responses”

Language and Grammar

The manuscript is well-written, but there are minor grammatical errors (e.g., "generalizability" vs. "generalisability"). Ensure consistency in spelling (American vs. British English).

Recommendation: Perform a thorough proofread to correct minor errors and ensure consistency.

Authors response: We thank the reviewer for this helpful suggestion. We have now performed a thorough proofread of the entire manuscript and have ensured consistency in spelling, punctuation, and grammar throughout

Statistical and Methodological Strengths

The use of both Cronbach’s alpha and McDonald’s omega to assess reliability is a strength, as it addresses limitations of alpha alone.

The inclusion of measurement invariance testing across gender and discipline adds robustness to the findings.

The cultural adaptation process (e.g., replacing "placement" with "department") is well-executed and contextually appropriate.

Authors response: We sincerely thank the reviewer for recognizing these methodological strengths.

The manuscript is methodologically sound and makes a valuable contribution to the field. The requested revisions are primarily technical (e.g., re-running factor analyses with EFA/Promax rotation) and clarificatory (e.g., expanding definitions and justifications). Once these issues are addressed, the manuscript will be suitable for publication.

Additional Recommendations for Future Work

Longitudinal Studies: Assess test-retest reliability and the tool's sensitivity to changes in the learning environment over time.

Cross-Cultural Validation: Extend validation to other LMICs to further establish the tool's global applicability.

Integration with Performance Metrics: Explore correlations between HEMLEM scores and objective clinical performance measures (e.g., OSCE results).

The authors are commended for their thorough work, and I look forward to seeing the revised manuscript.

Authors response: We sincerely thank the reviewer for this thoughtful summary and for recognizing the methodological soundness and overall contribution of our work. We have addressed all technical and clarificatory comments as outlined in our responses above, including justifications for factor analytic methods, enhanced definitions, expanded reliability discussion, and transparent data practices.

We also appreciate the reviewer’s recommendations for future directions, including longitudinal evaluation, cross-cultural validation in other LMICs, and potential correlations with objective clinical outcomes. We have retained and emphasized these in the Discussion section to guide future research. We are grateful for the encouraging feedback and constructive suggestions. We look forward to the acceptance of this paper.

Brown, T. A. (2015). Confirmatory factor analysis for applied research: Guilford publications.

Gunawan, J., Marzilli, C., & Aungsuroch, Y. (2021). Establishing appropriate sample size for developing and validating a questionnaire in nursing research. Belitung Nurs J, 7(5), 356-360. doi:10.33546/bnj.1927

Kline, R. B. (2023). Principles and practice of structural equation modeling: Guilford publications.

Ledesma, R. D., Ferrando, P. J., Trógolo, M. A., Poó, F. M., Tosi, J. D., & Castro, C. (2021). Exploratory factor analysis in transportation research: Current practices and recommendations. Transportation research part F: traffic psychology and behaviour, 78, 340-352.

McNeish, D., & Wolf, M. G. (2023). Dynamic fit index cutoffs for confirmatory factor analysis models. Psychological Methods, 28(1), 61.

Schreiber, J. B. (2021). Issues and recommendations for exploratory factor analysis and principal component analysis. Research in Social and Administrative Pharmacy, 17(5), 1004-1011.

Shrestha, N. (2021). Factor analysis as a tool for survey analysis. American journal of Applied Mathematics and statistics, 9(1), 4-11.

Widaman, K. F., & Helm, J. L. (2023). Exploratory factor analysis and confirmatory factor analysis.

---

## [Decision Letter · Decision Letter 3]

3 Sep 2025

Dear Dr. Sethi,

Thank you for submitting your manuscript to PLOS ONE. After careful consideration, we feel that it has merit but does not fully meet PLOS ONE’s publication criteria as it currently stands. Therefore, we invite you to submit a revised version of the manuscript that addresses the points raised during the review process.

We look forward to receiving your revised manuscript.

Kind regards,

Fatemeh Farshad

Guest Editor

PLOS ONE

Journal Requirements:

Additional Editor Comments (if provided):

Reviewer #5:

This manuscript describes the cultural adaptation and psychometric validation of the HEMLEM tool for use in Pakistani medical and dental schools. The study addresses an important gap in the literature concerning the assessment of clinical micro-learning environments in a new cultural context. The work is generally well-structured, and the authors have undertaken a rigorous process for content validation. The use of advanced statistical techniques like Multi-Group Confirmatory Factor Analysis (MGCFA) for measurement invariance is a significant strength. However, several critical methodological and statistical concerns, particularly regarding the conflation of Exploratory Factor Analysis (EFA) and Confirmatory Factor Analysis (CFA), undermine the validity of the reported findings. These issues require substantial revision.

Recommendation: Major Revision

Detailed Comments for the Authors

1. Major Conceptual and Statistical Flaw: Conflation of EFA and CFA

This is the most critical issue in the manuscript. The authors state that their primary analysis was CFA to test an a priori three-factor model. However, the methodology and results sections are heavily, and confusingly, reliant on EFA techniques.

Page 21, Line 293-303; Page 22, Table 3: The presentation of a "Rotated Component Matrix" using Principal Component Analysis (PCA) with Varimax rotation is a purely exploratory technique. PCA is a data reduction method, not a latent variable modeling technique like EFA or CFA. Presenting this after claiming to have run a CFA creates a contradictory narrative.

Justification in Response to Reviewers (Page 2-3): The authors' response to Reviewer 5, while defending the use of CFA, acknowledges that Table 3 displays "preliminary loadings for reader transparency." These exploratory values should not be the basis for assigning items to domains (as done on Lines 298-300: "based on the factor loading values, items 1 till 6 were assigned to domain 1..."). In a true CFA, the factor structure (which items load on which factor) is pre-specified based on theory or prior EFA, not determined from the current dataset.

Recommendation: The authors must clarify their analytical strategy. If an EFA/PCA was indeed conducted on the Pakistani dataset to explore the factor structure, this must be explicitly stated in the methods before the CFA is introduced. The results of the EFA should be presented first, followed by the CFA which tests the structure suggested by the EFA (or the original theoretical structure). The current presentation suggests a circular analysis where the same data is used to explore and then confirm the structure, which is methodologically unsound. If the CFA was truly theory-driven and confirmatory, the EFA/PCA results (Table 3) should be removed entirely, as they are redundant and misleading.

2. Sample Size Justification

Page 17, Lines 201-202: The justification ("10 participants per item") is outdated and not a robust methodological standard. While the final sample size of N=628 is excellent for a 12-item scale and provides high statistical power, the justification should be updated with contemporary references that consider desired factor loading magnitudes, number of factors, and anticipated communalities (e.g., Kyriazos, 2018; Wolf et al., 2013).

Page 25, Lines 390-399: The sample size discussion in the manuscript is better, citing Comrey & Lee and others. This stronger justification should be moved to the Methods section.

3. Measurement Invariance Reporting

Page 24, Lines 334-344: The MGCFA is a strength. However, the results are not sufficiently detailed. The authors must report the fit indices (χ², df, CFI, TLI, RMSEA, SRMR) for the configural, metric, and scalar models for both gender and discipline in the main text or a table, not just refer to supplemental materials. The critical values for evaluating invariance (ΔCFI < 0.01, ΔRMSEA < 0.015) should be stated in the Methods section.

4. Reliability Reporting

Page 24, Line 346-347: Reporting both Cronbach's alpha and McDonald's omega is a best practice. However, the value of "composite reliability" (0.766) is mentioned without a clear definition. It should be specified that this is likely the Maximal Reliability for the general factor (H) if referring to the bifactor model, or the construct reliability for the specific factors. The calculation formula (e.g., Raykov's rho) should be referenced.

5. Terminology and Clarity

Page 22, Lines 304-312: The distinction between "Autonomy" and "Teaching Quality" is helpful. However, the manuscript later reverts to the original three-factor model labels ("Supervision, Autonomy, Atmosphere"). There is an inconsistency between the two-domain structure mentioned in the EFA section (Staff Attitudes/Teaching Quality) and the three-domain CFA structure. This discrepancy must be explicitly addressed and resolved. The labels used in the final model must be consistent throughout the paper.

6. Data Analysis Description

Page 18, Lines 233-242; Page 24, Lines 334-344: The description of MGCFA is good. However, the software mention on Page 24, Line 245 (R with Lavaan) contradicts the earlier statement on Page 18, Line 213 (IBM SPSS AMOS). This must be corrected for consistency.

Page 18, Line 214: The assessment of normality via skewness and kurtosis is appropriate. It would be beneficial to state the specific cut-off values used (e.g., |skewness| < 2, |kurtosis| < 7).

7. Limitations

Page 27-28, Lines 480-497: The limitations section is comprehensive and well-written. It appropriately acknowledges the sampling method, lack of test-retest reliability, and potential for bias.

Summary of Required Revisions

Clarify the Factor Analysis Sequence: Decide and clearly report whether an EFA was conducted. If yes, present EFA results first, then use them to inform the CFA model. If the study was purely confirmatory, remove the EFA/PCA results (Table 3) and all references to using the current data to assign items to factors.

Resolve Terminology Inconsistencies: Ensure the names of the factors and subscales are consistent between the EFA (if kept) and CFA sections and throughout the manuscript.

Provide Detailed MGCFA Results: Report full fit indices for all invariance models in the main text.

Improve Sample Size Justification: Replace the "10 per item" rule with a more contemporary justification in the Methods section.

Correct Software Inconsistencies: Ensure the software used for each analysis is correctly and consistently reported.

Minor Proofreading: Address minor grammatical errors and ensure consistency in spelling (e.g., "generalizability" vs. "generalisability").

Conclusion

The authors have undertaken a valuable study with considerable methodological strengths, including a strong cultural adaptation process, a large sample, and the application of advanced statistical techniques like MGCFA. However, the fundamental confusion between exploratory and confirmatory analysis paradigms is a major flaw that affects the interpretation and validity of the core findings. Addressing this central issue is paramount. Once these major revisions are complete, this manuscript has the potential to be a significant contribution to the field of health professions education.

Reviewers' comments:

Reviewer's Responses to Questions

**Comments to the Author**

Reviewer #5: (No Response)

2. Is the manuscript technically sound, and do the data support the conclusions?

Reviewer #5: (No Response)

3. Has the statistical analysis been performed appropriately and rigorously?

Reviewer #5: (No Response)

4. Have the authors made all data underlying the findings in their manuscript fully available?

Reviewer #5: (No Response)

5. Is the manuscript presented in an intelligible fashion and written in standard English?

Reviewer #5: (No Response)

Reviewer #5: This manuscript describes the cultural adaptation and psychometric validation of the HEMLEM tool for use in Pakistani medical and dental schools. The study addresses an important gap in the literature concerning the assessment of clinical micro-learning environments in a new cultural context. The work is generally well-structured, and the authors have undertaken a rigorous process for content validation. The use of advanced statistical techniques like Multi-Group Confirmatory Factor Analysis (MGCFA) for measurement invariance is a significant strength. However, several critical methodological and statistical concerns, particularly regarding the conflation of Exploratory Factor Analysis (EFA) and Confirmatory Factor Analysis (CFA), undermine the validity of the reported findings. These issues require substantial revision.

Recommendation: Major Revision

Detailed Comments for the Authors

1. Major Conceptual and Statistical Flaw: Conflation of EFA and CFA

This is the most critical issue in the manuscript. The authors state that their primary analysis was CFA to test an a priori three-factor model. However, the methodology and results sections are heavily, and confusingly, reliant on EFA techniques.

Page 21, Line 293-303; Page 22, Table 3: The presentation of a "Rotated Component Matrix" using Principal Component Analysis (PCA) with Varimax rotation is a purely exploratory technique. PCA is a data reduction method, not a latent variable modeling technique like EFA or CFA. Presenting this after claiming to have run a CFA creates a contradictory narrative.

Justification in Response to Reviewers (Page 2-3): The authors' response to Reviewer 5, while defending the use of CFA, acknowledges that Table 3 displays "preliminary loadings for reader transparency." These exploratory values should not be the basis for assigning items to domains (as done on Lines 298-300: "based on the factor loading values, items 1 till 6 were assigned to domain 1..."). In a true CFA, the factor structure (which items load on which factor) is pre-specified based on theory or prior EFA, not determined from the current dataset.

Recommendation: The authors must clarify their analytical strategy. If an EFA/PCA was indeed conducted on the Pakistani dataset to explore the factor structure, this must be explicitly stated in the methods before the CFA is introduced. The results of the EFA should be presented first, followed by the CFA which tests the structure suggested by the EFA (or the original theoretical structure). The current presentation suggests a circular analysis where the same data is used to explore and then confirm the structure, which is methodologically unsound. If the CFA was truly theory-driven and confirmatory, the EFA/PCA results (Table 3) should be removed entirely, as they are redundant and misleading.

2. Sample Size Justification

Page 17, Lines 201-202: The justification ("10 participants per item") is outdated and not a robust methodological standard. While the final sample size of N=628 is excellent for a 12-item scale and provides high statistical power, the justification should be updated with contemporary references that consider desired factor loading magnitudes, number of factors, and anticipated communalities (e.g., Kyriazos, 2018; Wolf et al., 2013).

Page 25, Lines 390-399: The sample size discussion in the manuscript is better, citing Comrey & Lee and others. This stronger justification should be moved to the Methods section.

3. Measurement Invariance Reporting

Page 24, Lines 334-344: The MGCFA is a strength. However, the results are not sufficiently detailed. The authors must report the fit indices (χ², df, CFI, TLI, RMSEA, SRMR) for the configural, metric, and scalar models for both gender and discipline in the main text or a table, not just refer to supplemental materials. The critical values for evaluating invariance (ΔCFI < 0.01, ΔRMSEA < 0.015) should be stated in the Methods section.

4. Reliability Reporting

Page 24, Line 346-347: Reporting both Cronbach's alpha and McDonald's omega is a best practice. However, the value of "composite reliability" (0.766) is mentioned without a clear definition. It should be specified that this is likely the Maximal Reliability for the general factor (H) if referring to the bifactor model, or the construct reliability for the specific factors. The calculation formula (e.g., Raykov's rho) should be referenced.

5. Terminology and Clarity

Page 22, Lines 304-312: The distinction between "Autonomy" and "Teaching Quality" is helpful. However, the manuscript later reverts to the original three-factor model labels ("Supervision, Autonomy, Atmosphere"). There is an inconsistency between the two-domain structure mentioned in the EFA section (Staff Attitudes/Teaching Quality) and the three-domain CFA structure. This discrepancy must be explicitly addressed and resolved. The labels used in the final model must be consistent throughout the paper.

6. Data Analysis Description

Page 18, Lines 233-242; Page 24, Lines 334-344: The description of MGCFA is good. However, the software mention on Page 24, Line 245 (R with Lavaan) contradicts the earlier statement on Page 18, Line 213 (IBM SPSS AMOS). This must be corrected for consistency.

Page 18, Line 214: The assessment of normality via skewness and kurtosis is appropriate. It would be beneficial to state the specific cut-off values used (e.g., |skewness| < 2, |kurtosis| < 7).

7. Limitations

Page 27-28, Lines 480-497: The limitations section is comprehensive and well-written. It appropriately acknowledges the sampling method, lack of test-retest reliability, and potential for bias.

Summary of Required Revisions

Clarify the Factor Analysis Sequence: Decide and clearly report whether an EFA was conducted. If yes, present EFA results first, then use them to inform the CFA model. If the study was purely confirmatory, remove the EFA/PCA results (Table 3) and all references to using the current data to assign items to factors.

Resolve Terminology Inconsistencies: Ensure the names of the factors and subscales are consistent between the EFA (if kept) and CFA sections and throughout the manuscript.

Provide Detailed MGCFA Results: Report full fit indices for all invariance models in the main text.

Improve Sample Size Justification: Replace the "10 per item" rule with a more contemporary justification in the Methods section.

Correct Software Inconsistencies: Ensure the software used for each analysis is correctly and consistently reported.

Minor Proofreading: Address minor grammatical errors and ensure consistency in spelling (e.g., "generalizability" vs. "generalisability").

Conclusion

The authors have undertaken a valuable study with considerable methodological strengths, including a strong cultural adaptation process, a large sample, and the application of advanced statistical techniques like MGCFA. However, the fundamental confusion between exploratory and confirmatory analysis paradigms is a major flaw that affects the interpretation and validity of the core findings. Addressing this central issue is paramount. Once these major revisions are complete, this manuscript has the potential to be a significant contribution to the field of health professions education.

**Do you want your identity to be public for this peer review?** For information about this choice, including consent withdrawal, please see our Privacy Policy

Reviewer #5: No

---

## [Author Response · Author response to Decision Letter 4]

25 Sep 2025

Response to editor Manuscript ID PONE-D-24-54864R3 - EMID:2eee3f8f87e5fc7a entitled "Contextual validation of the HEMLEM tool used for measuring clinical micro-learning environments" submitted to PLOS ONE

Fatemeh Farshad

Guest Editor

PLOS ONE

Re: Manuscript ID PONE-D-24-54864R3 - EMID:2eee3f8f87e5fc7a

Many thanks for your careful consideration of our manuscript and thoughtful feedback. I have copied all the comments below from reviewer# 5 and provided a detailed explanation of how we have addressed them. We have provided a copy with highlighted changes and a clear version to reflect all the revisions. I am happy to make further revisions if you wish.

Best Regards

Ahsan Sethi (on behalf of the authors)

Reviewer 5 Comments

This manuscript describes the cultural adaptation and psychometric validation of the HEMLEM tool for use in Pakistani medical and dental schools. The study addresses an important gap in the literature concerning the assessment of clinical micro-learning environments in a new cultural context. The work is generally well-structured, and the authors have undertaken a rigorous process for content validation. The use of advanced statistical techniques like Multi-Group Confirmatory Factor Analysis (MGCFA) for measurement invariance is a significant strength. However, several critical methodological and statistical concerns, particularly regarding the conflation of Exploratory Factor Analysis (EFA) and Confirmatory Factor Analysis (CFA), undermine the validity of the reported findings. These issues require substantial revision.

Author response: Thank you. We have addressed all your comments in the revised manuscript.

Major Conceptual and Statistical Flaw: Conflation of EFA and CFA

This is the most critical issue in the manuscript. The authors state that their primary analysis was CFA to test an a priori three-factor model. However, the methodology and results sections are heavily, and confusingly, reliant on EFA techniques.

Page 21, Line 293-303; Page 22, Table 3: The presentation of a "Rotated Component Matrix" using Principal Component Analysis (PCA) with Varimax rotation is a purely exploratory technique. PCA is a data reduction method, not a latent variable modeling technique like EFA or CFA. Presenting this after claiming to have run a CFA creates a contradictory narrative.

Justification in Response to Reviewers (Page 2-3): The authors' response to Reviewer 5, while defending the use of CFA, acknowledges that Table 3 displays "preliminary loadings for reader transparency." These exploratory values should not be the basis for assigning items to domains (as done on Lines 298-300: "based on the factor loading values, items 1 till 6 were assigned to domain 1..."). In a true CFA, the factor structure (which items load on which factor) is pre-specified based on theory or prior EFA, not determined from the current dataset.

Recommendation: The authors must clarify their analytical strategy. If an EFA/PCA was indeed conducted on the Pakistani dataset to explore the factor structure, this must be explicitly stated in the methods before the CFA is introduced. The results of the EFA should be presented first, followed by the CFA which tests the structure suggested by the EFA (or the original theoretical structure). The current presentation suggests a circular analysis where the same data is used to explore and then confirm the structure, which is methodologically unsound. If the CFA was truly theory-driven and confirmatory, the EFA/PCA results (Table 3) should be removed entirely, as they are redundant and misleading.

Author response: We thank the reviewer for this critical and helpful observation. Based on this feedback, we have entirely removed Table 3 and all references to PCA or Exploratory analysis from both the Methods and Results sections. The current manuscript follows a strictly confirmatory approach, with a pre-specified three-factor model derived from the original theoretical structure of the HEMLEM instrument. All item-to-factor assignments were made a priori, and no data-driven reallocation was performed. The CFA was conducted solely to test the applicability of this established structure in a new cultural context. The revised Statistical Analyses and Results sections now reflect this strategy with added clarity, ensuring there is no conflation of exploratory and confirmatory methods.

2. Sample Size Justification

Page 17, Lines 201-202: The justification ("10 participants per item") is outdated and not a robust methodological standard. While the final sample size of N=628 is excellent for a 12-item scale and provides high statistical power, the justification should be updated with contemporary references that consider desired factor loading magnitudes, number of factors, and anticipated communalities (e.g., Kyriazos, 2018; Wolf et al., 2013).

Page 25, Lines 390-399: The sample size discussion in the manuscript is better, citing Comrey & Lee and others. This stronger justification should be moved to the Methods section.

Author response: We appreciate this insightful comment. The outdated "10 participants per item" rule has been removed from the Methods section. Instead, we have incorporated a more comprehensive justification that reflects current psychometric standards, including citations to Wolf et al. (2013) and Kyriazos (2023). The revised paragraph has been relocated to the ‘Sample Size’ subsection in the Methods section, as recommended. The paragraph in the ‘Discussion’ section has also been revised accordingly

3. Measurement Invariance Reporting

Page 24, Lines 334-344: The MGCFA is a strength. However, the results are not sufficiently detailed. The authors must report the fit indices (χ², df, CFI, TLI, RMSEA, SRMR) for the configural, metric, and scalar models for both gender and discipline in the main text or a table, not just refer to supplemental materials. The critical values for evaluating invariance (ΔCFI < 0.01, ΔRMSEA < 0.015) should be stated in the Methods section.

Author response: We thank the reviewer for this valuable observation.

We have added full model fit indices (χ², df, CFI, TLI, RMSEA, SRMR) for configural, metric, and scalar models across both gender and academic discipline groups.

These results are now presented within the main text and supported by two newly created tables (Tables 3 and 4) that clearly display the MGCFA outputs.

4. Reliability Reporting

Page 24, Line 346-347: Reporting both Cronbach's alpha and McDonald's omega is a best practice. However, the value of "composite reliability" (0.766) is mentioned without a clear definition. It should be specified that this is likely the Maximal Reliability for the general factor (H) if referring to the bifactor model, or the construct reliability for the specific factors. The calculation formula (e.g., Raykov's rho) should be referenced.

Author response: We thank the reviewer for this important observation regarding the reporting of composite reliability.

We have added a clear description of how composite reliability was calculated in the methods section.

We have corrected an error in our original calculation and updated the text in results section under the ‘Reliability’ subsection

We have also added a comprehensive table in the supplementary file (Table S7) showing all three reliability measures (Cronbach's α, McDonald's ω, and CR).

We note that the originally reported CR value of 0.766 was incorrectly calculated. The correct value based on CFA factor loadings is 0.901, which aligns better with the other reliability measures and indicates excellent construct reliability.

5. Terminology and Clarity

Page 22, Lines 304-312: The distinction between "Autonomy" and "Teaching Quality" is helpful. However, the manuscript later reverts to the original three-factor model labels ("Supervision, Autonomy, Atmosphere"). There is an inconsistency between the two-domain structure mentioned in the EFA section (Staff Attitudes/Teaching Quality) and the three-domain CFA structure. This discrepancy must be explicitly addressed and resolved. The labels used in the final model must be consistent throughout the paper.

Author response: We appreciate the reviewer’s observation regarding inconsistent domain labels. In response, we have removed the exploratory terminology ("Teaching Quality", "Staff Attitudes") that appeared in earlier drafts and now consistently refer to the original HEMLEM domains: Supervision, Autonomy, and Atmosphere.

6. Data Analysis Description

Page 18, Lines 233-242; Page 24, Lines 334-344: The description of MGCFA is good. However, the software mention on Page 24, Line 245 (R with Lavaan) contradicts the earlier statement on Page 18, Line 213 (IBM SPSS AMOS). This must be corrected for consistency.

Page 18, Line 214: The assessment of normality via skewness and kurtosis is appropriate. It would be beneficial to state the specific cut-off values used (e.g., |skewness| < 2, |kurtosis| < 7).

Author response: We thank the reviewer for identifying the inconsistency in the software description. We have revised the Methods section to correctly state that CFA and MGCFA were conducted using R with the Lavaan package, while descriptive statistics were performed using SPSS.

We have also clarified the criteria for assessing normality by specifying that acceptable skewness and kurtosis values were set at < 2 and < 7, respectively, based on standard guidelines.

7. Limitations

Page 27-28, Lines 480-497: The limitations section is comprehensive and well-written. It appropriately acknowledges the sampling method, lack of test-retest reliability, and potential for bias.

Summary of Required Revisions

Clarify the Factor Analysis Sequence: Decide and clearly report whether an EFA was conducted. If yes, present EFA results first, then use them to inform the CFA model. If the study was purely confirmatory, remove the EFA/PCA results (Table 3) and all references to using the current data to assign items to factors.

Resolve Terminology Inconsistencies: Ensure the names of the factors and subscales are consistent between the EFA (if kept) and CFA sections and throughout the manuscript.

Provide Detailed MGCFA Results: Report full fit indices for all invariance models in the main text.

Improve Sample Size Justification: Replace the "10 per item" rule with a more contemporary justification in the Methods section.

Correct Software Inconsistencies: Ensure the software used for each analysis is correctly and consistently reported.

Minor Proofreading: Address minor grammatical errors and ensure consistency in spelling (e.g., "generalizability" vs. "generalisability").

Conclusion

The authors have undertaken a valuable study with considerable methodological strengths, including a strong cultural adaptation process, a large sample, and the application of advanced statistical techniques like MGCFA. However, the fundamental confusion between exploratory and confirmatory analysis paradigms is a major flaw that affects the interpretation and validity of the core findings. Addressing this central issue is paramount. Once these major revisions are complete, this manuscript has the potential to be a significant contribution to the field of health professions education.

Author response: We thank the reviewer for their thorough and constructive feedback. We have carefully addressed each of the concerns noted in the summary and the conclusion

---

## [Decision Letter · Decision Letter 4]

27 Oct 2025

Dear Dr. Sethi,

Thank you for submitting your manuscript to PLOS ONE. After careful consideration, we feel that it has merit but does not fully meet PLOS ONE’s publication criteria as it currently stands. Therefore, we invite you to submit a revised version of the manuscript that addresses the points raised during the review process.

https://journals.plos.org/plosone/s/submission-guidelines#loc-laboratory-protocols . Additionally, PLOS ONE offers an option for publishing peer-reviewed Lab Protocol articles, which describe protocols hosted on protocols.io. Read more information on sharing protocols at https://plos.org/protocols?utm_medium=editorial-email&utm_source=authorletters&utm_campaign=protocols .

We look forward to receiving your revised manuscript.

Kind regards,

Fatemeh Farshad

Guest Editor

PLOS ONE

**Journal Requirements:**

**Additional Editor Comments:**

Due to the reviewer's comment It is recommended to revise by an English professional editor .

Reviewers' comments:

Reviewer's Responses to Questions

**Comments to the Author**

Reviewer #6: All comments have been addressed

2. Is the manuscript technically sound, and do the data support the conclusions?

Reviewer #6: Yes

3. Has the statistical analysis been performed appropriately and rigorously?

Reviewer #6: Yes

4. Have the authors made all data underlying the findings in their manuscript fully available?

Reviewer #6: Yes

5. Is the manuscript presented in an intelligible fashion and written in standard English?

Reviewer #6: No

**Reviewer #6:**  It is a good topic . After 4 rounds and review , it has been better .

It is recommended to revise by an English professional editor . ( Minor revision).

**Do you want your identity to be public for this peer review?**  For information about this choice, including consent withdrawal, please see our Privacy Policy

Reviewer #6: No

---

## [Author Response · Author response to Decision Letter 5]

10 Nov 2025

Thank you for your message. Unfortunately, we do not have any funds allocated for this project or for professional English editing services. Our team has extensive publication experience and a strong command of the English language.

After five rounds of revisions and feedback from six reviewers, we have carefully re-examined the manuscript and believe that the English is appropriate for publication. Unless there are specific areas that require further language improvement, we would kindly request that the paper be considered for acceptance at this stage.

---

## [Editor Report · Decision Letter 5]

12 Nov 2025

Contextual validation of HEMLEM tool used for measuring clinical micro-learning environments

PONE-D-24-54864R5

Dear Dr. Sethi,

We’re pleased to inform you that your manuscript has been judged scientifically suitable for publication and will be formally accepted for publication once it meets all outstanding technical requirements.

Kind regards,

Fatemeh Farshad

Guest Editor

PLOS ONE
---

## [Editor Report · Acceptance letter]

PONE-D-24-54864R5

PLOS ONE

Dear Dr. Sethi,

I'm pleased to inform you that your manuscript has been deemed suitable for publication in PLOS ONE. Congratulations! Your manuscript is now being handed over to our production team.

Kind regards,

on behalf of

Dr. Fatemeh Farshad

Guest Editor

PLOS ONE